# Dilated cardiomyopathy-associated RNA-binding motif protein 20 regulates long pre-mRNAs in neurons

**Giulia Di Bartolomei[1], Raúl Ortiz[1], Dietmar Schreiner[1], Susanne Falkner[1], Esther EJM Creemers[2], Peter Scheiffele[1]***

[1]Biozentrum, University of Basel, Basel, Switzerland; [2]Department of Experimental Cardiology, Amsterdam Cardiovascular Sciences, Amsterdam UMC, University of Amsterdam, Amsterdam, Netherlands

## eLife Assessment

This study reports that the RNA binding and cardiomyopathy-associated protein RBM20 is expressed in specific populations of neurons in the CNS, where it binds to and regulates the expression of synapse-related RNAs. This is an **important** finding because it reveals a new mechanism for gene regulation in neurons by an RNA binding protein previously studied in the heart; the authors also provide data to suggest that the mechanism by which RBM20 acts in neurons may be distinct from the splicing regulation studied in cardiac tissue. The data in support of the binding and regulation of RNAs by RBM20 is **compelling**, using leading edge sequencing methods to determine RNA binding profiles, and cell type specific genetics for evaluation of function.

***For correspondence:**
peter.scheiffele@unibas.ch

**Competing interest:** The authors declare that no competing interests exist.

**Abstract** Precise coordination of molecular programs and neuronal growth governs the formation, maintenance, and adaptation of neuronal circuits. RNA metabolism has emerged as a key regulatory node of neural development and nervous system pathologies. To uncover cell-type-specific RNA regulators, we systematically investigated expression of RNA recognition motif-containing proteins in the mouse neocortex. Surprisingly, we found RNA-binding motif protein 20 (RBM20), an alternative splicing regulator associated with dilated cardiomyopathy, to be expressed in cortical parvalbumin interneurons and mitral cells of the olfactory bulb. Genome-wide mapping of RBM20 target mRNAs revealed that neuronal RBM20 binds pre-mRNAs in distal intronic regions. Loss of neuronal RBM20 has only modest impact on alternative splice isoforms but results in a significant reduction in an array of mature mRNAs in the neuronal cytoplasm. This phenotype is particularly pronounced for genes with long introns that encode synaptic proteins. We hypothesize that RBM20 ensures fidelity of pre-mRNA splicing by suppressing nonproductive splicing events in long neuronal genes. This work highlights a common requirement for RBM20-dependent transcriptome regulation in cardiomyocytes and neurons and demonstrates that a major genetic risk factor of heart disease impacts neuronal gene expression.

## Introduction

Neurons exhibit complex transcriptional programs that instruct the specification of functionally distinct neuronal cell types. Functional and anatomical properties of neurons emerge during development. However, the molecular underpinnings that define cell-type-specific properties are only beginning to be elucidated. RNA-binding proteins (RBPs) have emerged as key regulators of neuronal function through modification of mRNA processing, localization, stability, and translation (*Ule and Darnell,*

*2006*; *Babitzke et al., 2009*; *Vuong et al., 2016*; *Mauger and Scheiffele, 2017*; *Holt et al., 2019*; *Ule and Blencowe, 2019*; *Gomez et al., 2021*). Moreover, RBP dysfunction is a significant contributor to pathologies, including neurodevelopmental and neurodegenerative conditions (*Ling et al., 2013*; *Lopez Soto et al., 2019*; *Gebauer et al., 2021*; *Schieweck et al., 2021*). Here, we discovered an unexpected neuronal function for the SR-related protein, RNA-binding motif protein 20 (RBM20). Thus far, RBM20 was considered to be muscle-specific and to represent a key alternative splicing regulator in cardiomyocytes. Mutations in the *RBM20* gene are linked to an aggressive form of dilated cardiomyopathy (*Brauch et al., 2009*; *Parikh et al., 2019*). In cardiomyocytes, RBM20 protein controls alternative exon usage of transcripts encoding key sarcomere components (Titin and Tropomyosin) and proteins involved in calcium signaling, such as CAMK2D and the α1 subunit of the L-type voltage-gated calcium channel (CACNA1C) (*Guo et al., 2012*; *Maatz et al., 2014*; *van den Hoogenhof et al., 2018*; *Zhu et al., 2021*). RBM20 contains an RNA recognition motif (RRM) domain, two zinc finger motifs, and an extended arginine/serine-rich region. This domain organization is shared with the paralogues Matrin-3 (MATR3) and ZNF638 (*Watanabe et al., 2018*) and is similar to FUS and TDP43, two RNA/DNA-binding proteins that regulate various steps of RNA metabolism. Comprehensive RBM20 protein-RNA interaction maps have been defined for cardiomyocytes (*Maatz et al., 2014*; *van den Hoogenhof et al., 2018*; *Briganti et al., 2020*). However, RBM20 expression and function in the brain remain unknown. Considering its highly selective expression in specific cell populations and the critical roles in cardiomyocytes, we hypothesized that neuronal RBM20 controls key steps of RNA metabolism and contributes to the regulation of neuronal gene expression.

## Results

### RBM20 is selectively expressed in specific GABAergic and glutamatergic neurons

To discover regulators of neuronal cell-type-specific transcriptomes, we analyzed a neuronal gene expression dataset covering cortical pyramidal cells and the major GABAergic interneuron populations of the mouse cortex (*Furlanis et al., 2019*). We generated a hand-curated list of 234 potential RBPs, selected based on the presence of at least one predicted RRM domain (*Figure 1A*, *Figure 1—figure supplement 1A*). Transcripts for 182 of these RRM proteins were significantly expressed in mouse cortical neurons. While some (such as *Eif3g*, *Hnrnpl*) were broadly detected across all neuronal populations, others were differentially expressed in specific neuron classes. Thus, *Rbm38* was almost exclusively expressed in VIP+ (vasoactive intestinal polypeptide-positive) interneurons and *Ppargc1a* and *Rbm20* in parvalbumin-positive (PV+) interneurons, indicating that they may play an important role in cell-type-specific gene regulation. Identification of *Rbm20* expression in the brain was surprising considering that the RBM20 protein was thought to be muscle-specific. Thus, we focused our further studies on probing potential functions of RBM20 in the mouse brain.

To validate the neuronal expression, we investigated *Rbm20* mRNA distribution in the mouse neocortex by RNA fluorescent in situ hybridization (FISH). This analysis confirmed significant expression of *Rbm20* in PV+ GABAergic interneurons (*Figure 1—figure supplement 1B–D*). Moreover, we detected *Rbm20* mRNA in somatostatin-positive (SST+) interneurons of the somatosensory cortex (*Figure 1—figure supplement 1B–D*). A survey of open-source datasets (*Sjöstedt et al., 2020*) revealed that *Rbm20* mRNA expression in PV+ interneurons is evolutionary conserved in human. To examine RBM20 at the protein level, we raised polyclonal antibodies and confirmed RBM20 expression in PV+ interneurons of the somatosensory cortex (*Figure 1B and C*). The parvalbumin-negative RBM20+ cells most likely represent somatostatin+-interneurons which also express significant levels of Rbm20 mRNA (*Figure 1—figure supplement 1C and D*). Notably, these experiments also uncovered neuronal RBM20 expression outside of the neocortex, with particularly high protein levels in the olfactory bulb (*Figure 1D and E*, see *Figure 1—figure supplement 1E* for comparison to protein expression in the heart and *Figure 3—figure supplement 1A and B* for antibody validation in immunostaining). In the olfactory bulb, RBM20 was restricted to glutamatergic cells in the mitral cell layer (MCL) and glomerular layer (GL) as opposed to the expression in GABAergic populations in the somatosensory cortex (*Figure 1D*). Genetic marking and in situ hybridization for markers *Slc17a6* and *Tbr2* confirmed the glutamatergic neuron expression of *Rbm20* mRNA in tufted and mitral cells (*Figure 1—figure supplement 2A–C*), with only very low levels in a small number of GABAergic

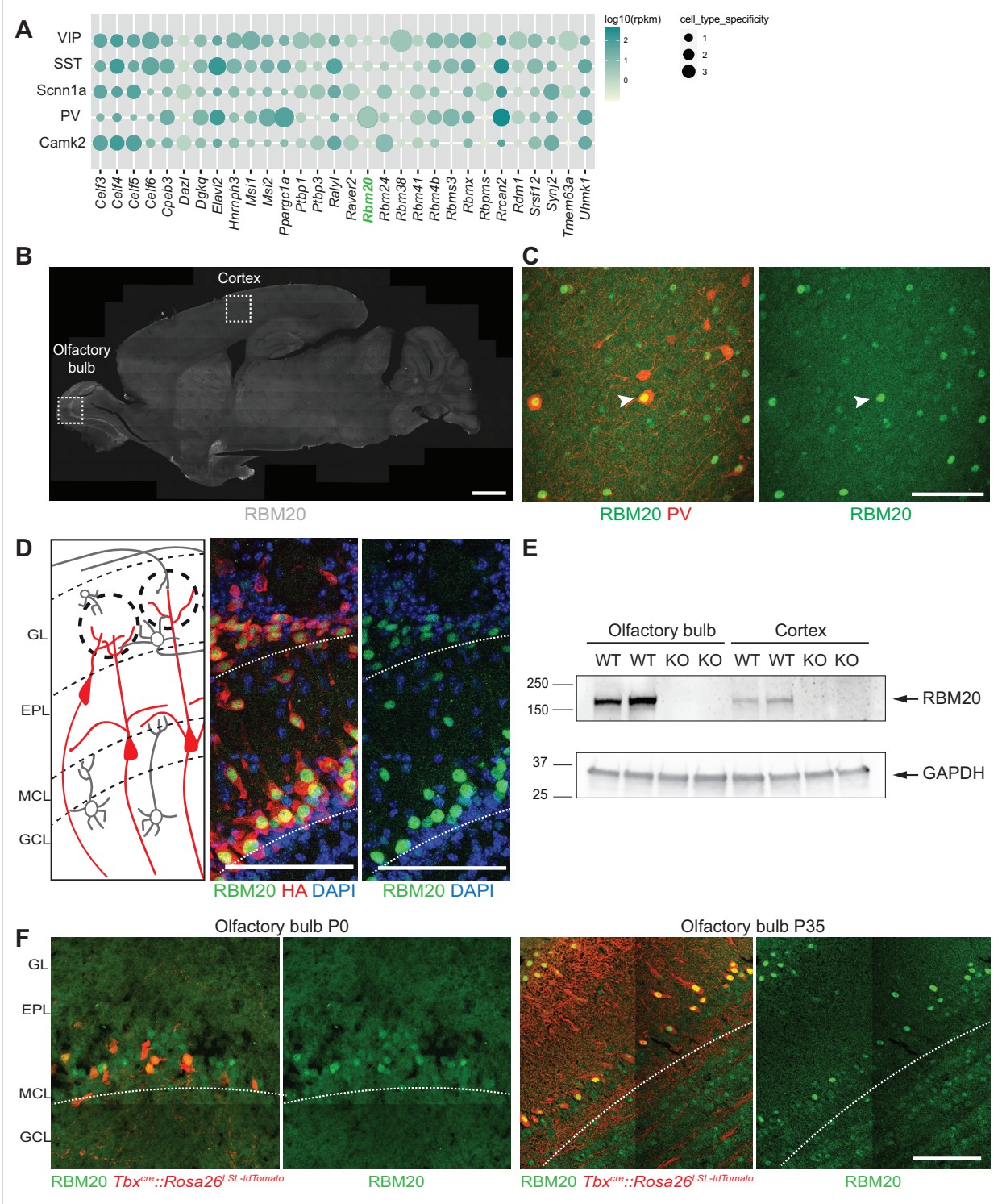

**Figure 1.** Characterization of *Rbm20* expression in the brain. (**A**) Dot plot of the expression of a hand-curated list of RNA-binding proteins (RBPs) across different neuronal neocortical populations. RBPs were chosen based on the presence of an RNA recognition motif (RRM) in their sequence and on the ranking of their gini-index value (only the first 20 RBPs displaying the highest gini-index value are displayed [see Materials and methods]). RBPs' expression was measured by Ribo-TRAP sequencing and expressed as RPKM values normalized over the mean expression across different neuronal populations. (**B**) Sagittal section of the mouse brain used for immunohistochemistry of RBM20 (gray). The somatosensory cortex and the olfactory bulb regions, where RBM20 is expressed, are highlighted. Scale bar: 1 mm. (**C**) Immunohistochemistry of RBM20 expression (green) in Parvalbumin-positive interneurons (red) in the neocortex. (**D**) Schematic illustration of the olfactory bulb circuitry and cell types (left). GL: glomerular layer, EPL: external plexiform layer, MCL: mitral cell layer, GCL: granule cell layer (left). RBM20 expression (green) is specific to the mitral cell layer and glomeruli layer of the

*Figure 1 continued on next page*

eLife Research article

Cell Biology | Neuroscience

olfactory bulb identified in the *Slc17a6^Cre^:: Rpl22^HA^* mouse line (HA staining in red) (middle and right). Scale bar: 100 μm. (**E**) Western blot probing RBM20 expression in olfactory bulb and cortex samples of wild-type (WT) and constitutive *Rbm20^KO^* mice (KO). The RBM20 band at ca. 150 kDa indicated with an arrow is selectively lost in KO tissue. GAPDH detection is used as loading control. For better visualization of the two proteins, the same tissue lysates were run on a 7.5% acrylamide gel (for RBM20 detection) and 4–20% acrylamide gradient gel (GAPDH). (**F**) Immunofluorescence of RBM20 (green) expression in the mitral cell layer (MCL) of P0 and P35 *Tbx21^Cre^:: Rosa26^LSL-tdTomato^* mice. Mitral cells and tufted neurons are labeled with the tdTomato reporter (red). Scale bar: 100 μm.

The online version of this article includes the following source data and figure supplement(s) for figure 1:

**Source data 1.** PDF file containing original western blots for *Figure 1E*, indicating the relevant bands and conditions.

**Source data 2.** Original files for western blot analysis displayed in *Figure 1E*.

**Figure supplement 1.** *Rbm20* mRNA expression in mouse neocortex.

**Figure supplement 1—source data 1.** PDF file containing original western blots for *Figure 1—figure supplement 1E*, indicating the relevant bands and conditions.

**Figure supplement 1—source data 2.** Original files for western blot analysis displayed in *Figure 1—figure supplement 1E*.

**Figure supplement 2.** *Rbm20* mRNA expression in mouse olfactory bulb.

neurons (*Figure 1—figure supplement 2D*). Similarly, we detected RBM20 protein in mitral cells of newborn mice (genetically marked with *Tbx^Cre^::Rosa26^LSL-tdTomato^*), suggesting that its expression is initiated during development and continues into the adult (*Figure 1F*).

## Neuronal RBM20 binds distal intronic regions of mRNAs encoding synaptic proteins

In cardiomyocytes, RBM20 acts as an alternative splicing regulator that gates exon inclusion by binding to exon-proximal intronic splicing silencers (*Li et al., 2013*; *Maatz et al., 2014*; *Dauksaite and Gotthardt, 2018*). Within the nucleus, RBM20 localizes to foci in close proximity to *Titin* transcripts (*Guo et al., 2012*; *Li et al., 2013*) and has been proposed to form 'splicing factories', sites for coordinated processing of mRNAs derived from multiple genes (*Bertero et al., 2019*). By contrast, neuronal RBM20 did not concentrate in nuclear foci but instead was distributed throughout the nucleoplasm (*Figure 2A*). This difference in the subnuclear localization likely arises from the lack of *Titin* and/or other similarly abundant mRNA targets in neuronal cells. To contrast neuronal and cardiomyocyte RNA targets, we mapped transcripts directly bound by RBM20 in the olfactory bulb and in the heart using cross-linking immunoprecipitation followed by sequencing (CLIP-seq) analysis (*Ule et al., 2003*; *Van Nostrand et al., 2017a*). Our RBM20 antibodies lacked sufficient affinity for immunoprecipitation. Thus, we generated an *Rbm20^HA^* knock-in mouse line where the endogenous RBM20 protein is tagged with a Histidine-Biotin acceptor peptide and a triple HA epitope (*Figure 2B and C*, *Figure 2—figure supplement 1A and B*). Introduction of this tag did not alter overall RBM20 protein levels or expression pattern (*Figure 2—figure supplement 1B and C*).

RBM20 CLIP-seq analysis on heart and olfactory bulb tissues from P35 to P40 *Rbm20^HA^* knock-in mice identified significant peaks in 956 and 2707 unique transcripts, respectively. Recovery of protein-RNA adducts was UV-cross-linking-dependent (*Figure 3—figure supplement 1A and B*), and CLIP tags obtained in biological replicates were highly correlated (two replicates for heart and three for olfactory bulb, *Figure 3—figure supplement 1C and D–F*). In both tissues, the majority of the identified RBM20 CLIP peaks mapped to introns (80% in the heart and 94% in the olfactory bulb) (*Figure 2D and E*). Only a low fraction of peaks was identified in the 3' untranslated region (11% in the heart and 1% in the olfactory bulb) and coding regions (7% in the heart and 3% in the olfactory bulb). RBM20 targets detected in the heart recapitulated all major target mRNAs identified in previous studies, including *Ttn, Tpm, Pdlim5, Scn5a, Camk2d, Cacna1c, Cacna1d* (*Guo et al., 2012*; *Maatz et al., 2014*; *van den Hoogenhof et al., 2018*; *Watanabe et al., 2018*; *Fenix et al., 2021*). In heart and olfactory bulb, approximately 90% of the intronic RBM20-binding sites localized to distal regions (>500 bp from splice site) (*Figure 2F*). The previously reported UCUU motif (*Guo et al., 2012*; *Upadhyay and Mackereth, 2020*) was significantly enriched at cross-link-induced truncation sites (CITS) (*Figure 3—figure supplement 1G*) and de novo binding motif enrichment analysis identified a corresponding U-rich motif at RBM20-bound sites (*Figure 3—figure supplement 1H*).

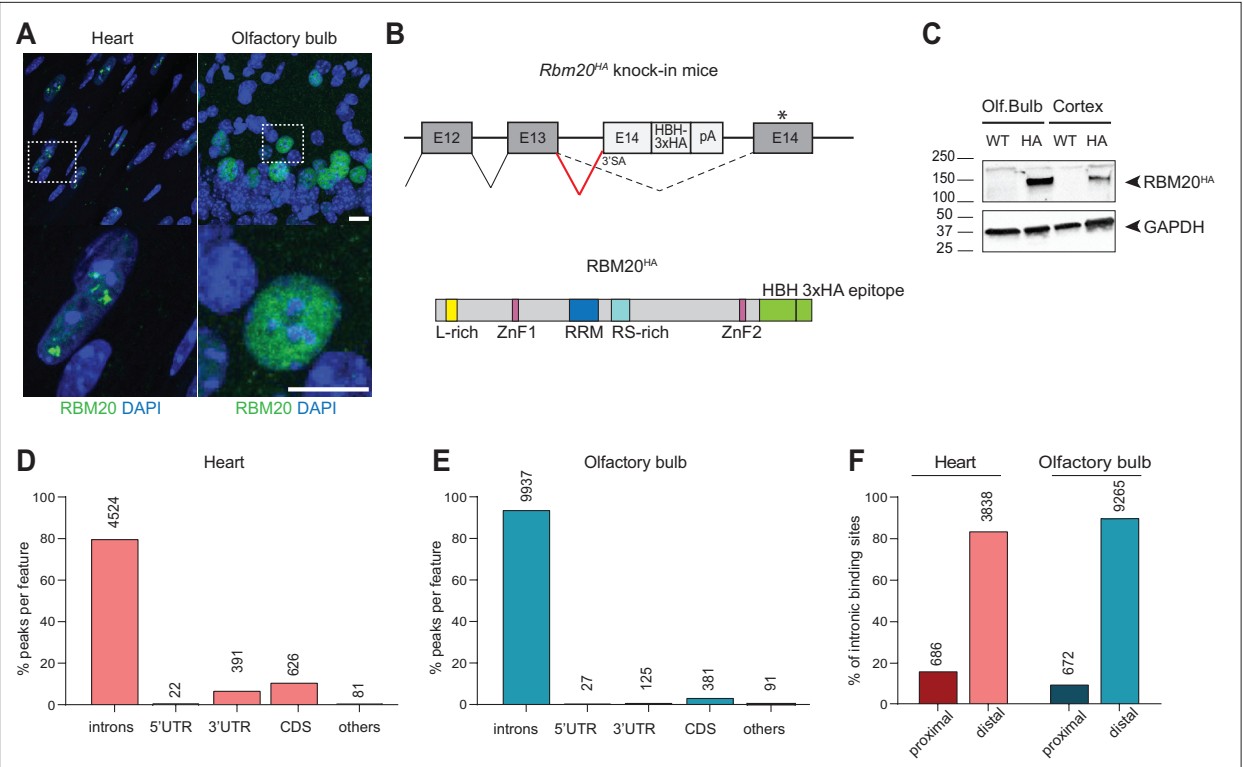

**Figure 2.** Identification of RBM20 direct targets in the heart and olfactory bulb. (**A**) Subnuclear localization of RBM20 (green) in heart cardiomyocytes (left) and mitral cells of the olfactory bulb (right) of wild-type (WT) mice at P35. Scale bar: 10 μm. (**B**) Schematic illustration of HA epitope tagging of endogenous RBM20 in mice. A cassette was inserted into the *Rbm20* locus containing a strong synthetic 3′ splicing acceptor site (3′SA) introduced into sequences derived from *Rbm20* exon 14 and in-frame fusion of a histidine-biotin-histidine-3xHA tag, followed by a polyadenylation signal (up). Schematic representation of the resulting RBM20^HA protein where the last exon of the protein is fused to a histidine-biotin-histidine-3xHA tag (down). (**C**) Western blot showing the validation of RBM20 expression in the olfactory bulb and cortex tissues of WT and *Rbm20^HA* tagged mice. GAPDH is used as a loading control. (**D**) Quantification of the percentage of peaks identified in the heart (red) and (**E**) olfactory bulb (blue) tissue in each genomic feature: introns, untranslated regions (3′UTR, 5′UTR), coding sequence (CDS), others (promoters, intergenic regions, noncoding regions). The absolute number of the peaks identified in each genomic feature is reported on top of the corresponding bar. (**F**) Bar plot showing the percentage of peaks identified in distal (>500 bp) or proximal (<500 bp) intronic regions in the heart (red) and olfactory bulb (blue). The absolute number of the peaks identified is reported on top of the corresponding bar.

The online version of this article includes the following source data and figure supplement(s) for figure 2:

**Source data 1.** PDF file containing original western blots for *Figure 2C*, indicating the relevant bands and conditions.

**Source data 2.** Original files for western blot analysis displayed in *Figure 2C*.

**Figure supplement 1.** Generation of *Rbm20* – HA-tagged mouse line.

When directly comparing heart and olfactory bulb CLIP-seq datasets, we found transcripts from 363 genes that were commonly bound by RBM20 in both tissues. Other transcripts like *Ttn* are recovered as RBM20-bound only in one of the two tissues due to tissue-specific expression of the mRNA (*Figure 3A*). Interestingly, *Camk2d* and *Cacna1c*, two important targets of RBM20 alternative splicing regulation in the heart, were identified as RBM20-bound in the olfactory bulb; however, binding sites mapped to different introns (*Figure 3A* and see *Supplementary file 1*). Finally, there was a sizable portion of pre-mRNAs (from 2721 genes) commonly expressed in both heart and olfactory bulb but selectively recovered as RBM20-bound in only one of the two tissues (539 genes in the heart and 2182 in the olfactory bulb). This suggests that RBM20 binds to target mRNAs in a tissue-specific manner. A tissue-specific function of RBM20 was further supported by gene ontology (GO) analysis. RBM20 target mRNAs in the heart showed enrichment for terms of muscle fiber components (M-band, Z-disc, T-tubule, *Figure 3B*). By contrast, target genes in the olfactory bulb showed enrichment for terms related to pre- and postsynaptic structures, ion channels, and cytoskeletal components (*Figure 3C* and *Supplementary file 2*). These include gephyrin (*Gphn*), a major scaffolding protein of GABAergic

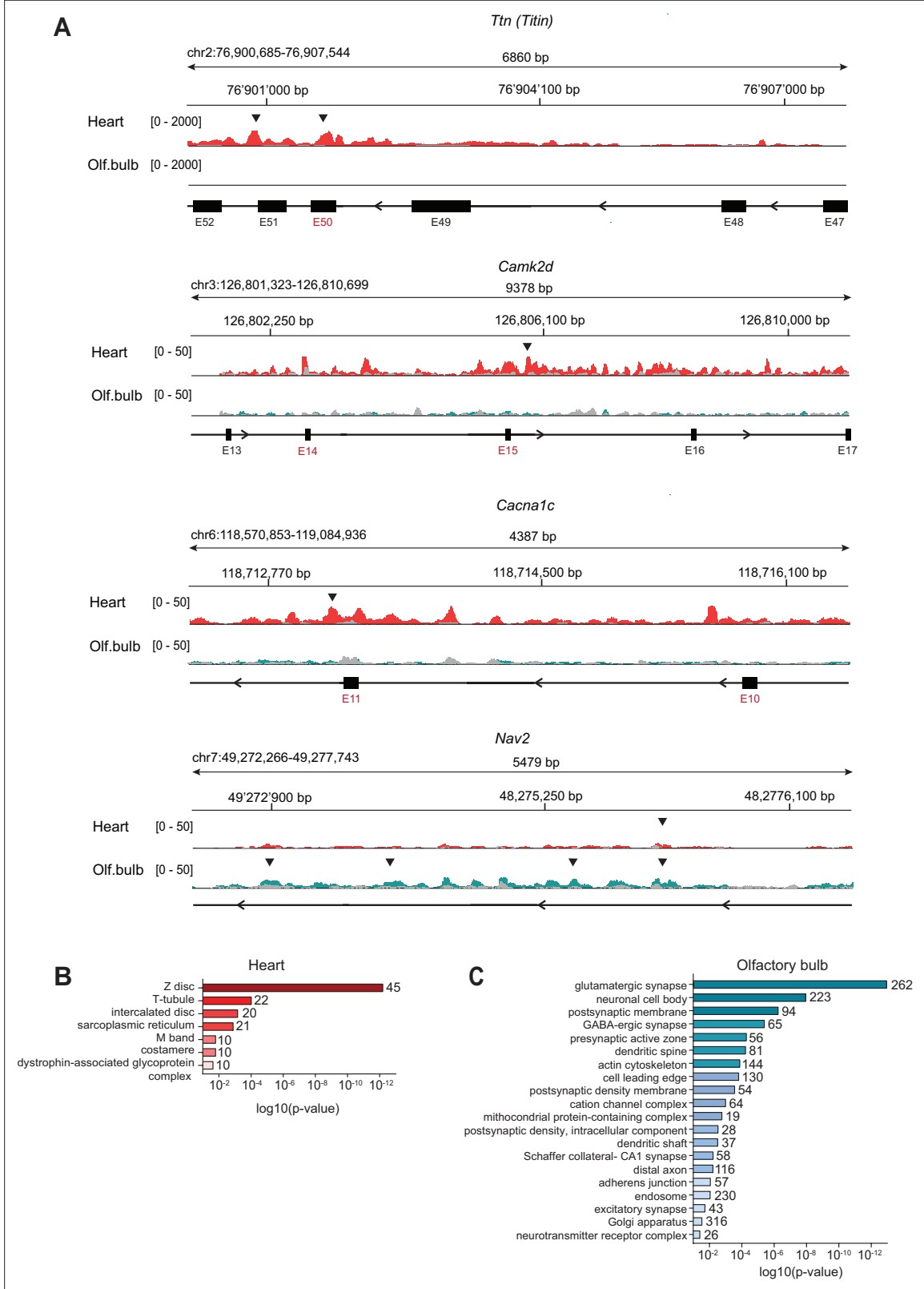

**Figure 3.** Identification of RBM20 direct mRNA targets in the heart and olfactory bulb. (**A**) Tracks illustrating RBM20 CLIP-seq signal for *Ttn, CamkIId, Cacna1c,* and *Nav2*. Read density obtained for heart samples (red traces) and olfactory bulb (green traces) with the corresponding input samples (overlaid traces in gray). CLIP peaks, considered statistically significant by irreproducible discovery rate (IDR) (score>540, equivalent to IDR<0.05), are marked by black arrowheads. RBM20-dependent alternative exons previously reported for cardiomyocytes are labeled in red. Note that in the olfactory

*Figure 3 continued on next page*

*Figure 3 continued*

bulb, RBM20-binding sites are identified on *CamkIId* and *Cacna1c* pre-mRNAs. However, these binding sites are distal (>500 bp) to the alternative exons. See **Supplementary file 1** for coordinates. (**B**) Illustration of the gene ontology (GO) categories of RBM20 mRNA targets in the heart (IDR<0.05). Cellular component analysis with Bonferroni correction (p-value≤0.05). The number of genes found in each category is displayed on top of each bar. Minimal number of genes identified in each category: five genes. (**C**) Illustration of the GO categories of RBM20 mRNA targets in the olfactory bulb (IDR<0.05). Cellular component analysis with Bonferroni correction (p-value≤0.05). The number of genes found in each category is displayed on top of each bar. Minimal number of genes identified in each category: five genes.

The online version of this article includes the following source data and figure supplement(s) for figure 3:

**Figure supplement 1.** Identification of RBM20-binding sites on transcript mRNAs.

**Figure supplement 1—source data 1.** PDF file containing original western blots and agarose gel of libraries for *Figure 3—figure supplement 1A and B*, indicating the relevant bands and conditions.

**Figure supplement 1—source data 2.** Original files for western blot and agarose gel of libraries displayed in *Figure 3—figure supplement 1A and B*.

synapses, the adhesion molecule Kirrel3, and Down syndrome cell adhesion molecule-like 1 (*Dscaml1*). This suggests that neuronal RBM20 plays a role in the regulation of synapse-related mRNA transcripts.

## Neuronal RBM20 is required for normal expression of long pre-mRNAs encoding synaptic proteins

To investigate the impact of RBM20 on the neuronal transcriptome, we performed loss-of-function experiments in *Rbm20* global and conditional knockout mice. *Rbm20* was conditionally inactivated selectively in either Parvalbumin-positive interneurons (*Pvalb^Cre^::Rbm20^fl/fl^*, referred to as '*Rbm20ΔPV*') or glutamatergic neurons (*Slc17a6^Cre^::Rbm20^fl/fl^*, referred to as '*Rbm20ΔSlc17a6* '). In global *Rbm20* knockout mice, RBM20 immune reactivity was abolished in the olfactory bulb and cortex (**Figure 1E** for western blots). Similarly, in the conditional knockout (cKO) mice, we observed a loss of RBM20 immune reactivity in the respective cell populations (**Figure 4—figure supplement 1A and B**). Importantly, neuronal cell types were normally specified in *Rbm20ΔSlc17a6* mice (**Figure 4—figure supplement 1C–F**). We then applied a Translating Ribosome Affinity Purification protocol (RiboTRAP) optimized for small tissue samples (**Heiman et al., 2014**; **Sanz et al., 2019**; **Di Bartolomei and Scheifele, 2022**) to uncover the impact of RBM20 loss-of-function on the neuronal transcriptome. Isolations were performed for *Rbm20ΔPV* and *Rbm20ΔSlc17a6* conditional knockout and matching *Rbm20^WT^* mice (postnatal days 35–40). For quality control, we confirmed appropriate enrichment and de-enrichment of cell-type-specific markers (**Figure 4—figure supplement 2A and B**). Subsequently, we assessed the transcriptomes by deep RNA-sequencing (4–5 replicates per condition, male and female mice, paired-end, 150 base pair reads, >80 million reads per replicate, see **Supplementary file 3** and **Figure 4—figure supplement 1C–F** for details). To identify genes with altered expression in *Rbm20* conditional knockout mice, we used DESEQ2 (**Love et al., 2014**). Using this approach, we did not identify significant differences in the overall transcriptome of *Rbm20ΔPV*, PV+ interneurons as compared to wild-type (WT) (**Figure 4A** and **Supplementary file 4**). However, in the olfactory bulb glutamatergic cells isolated from Rbm20ΔSlc17a6 mice, we identified deregulation of 409 genes (FC ≥1.5, adjusted p-value<0.01). 256 of these genes showed decreased expression in the cKO mice as opposed to transcripts from 153 transcripts that were elevated (**Figure 4B**). In the heart, RBM20 is considered a major regulator of alternative splicing. Thus, we quantified shifts in alternative exon usage upon *Rbm20* loss-of-function. We identified 859 differentially regulated alternative exons in PV+ interneurons and 1924 exons in the *Slc17a6+* population in the olfactory bulb (FC in splicing index ≥1.5, p-value≤0.05) (**Figure 4C and D** and **Supplementary file 5**). The functions of the gene products with alternative splicing deregulation were diverse, and the only significantly enriched GO term for genes with deregulated exons was 'mitochondrial protein-containing complexes' (p-value=0.0003, 100 genes identified) for the olfactory bulb (*Slc17a6+*) population (**Supplementary file 6**).

To uncover transcript isoforms directly regulated by RBM20, we probed the intersection of the RBM20-binding sites identified by CLIP and alternative exon expression in the *Slc17a6+* cell population. Of the 1924 exons with differential incorporation in *Rbm20ΔSlc17a6* cells, there were 659 exons with at least one significant RBM20 CLIP peak mapping to the corresponding gene. This frequency was higher than expected for a random distribution of CLIP peaks over all genes expressed in the sample (p=5.2 * 10^{-8}, hypergeometric distribution test for enrichment analysis). Interestingly, the vast

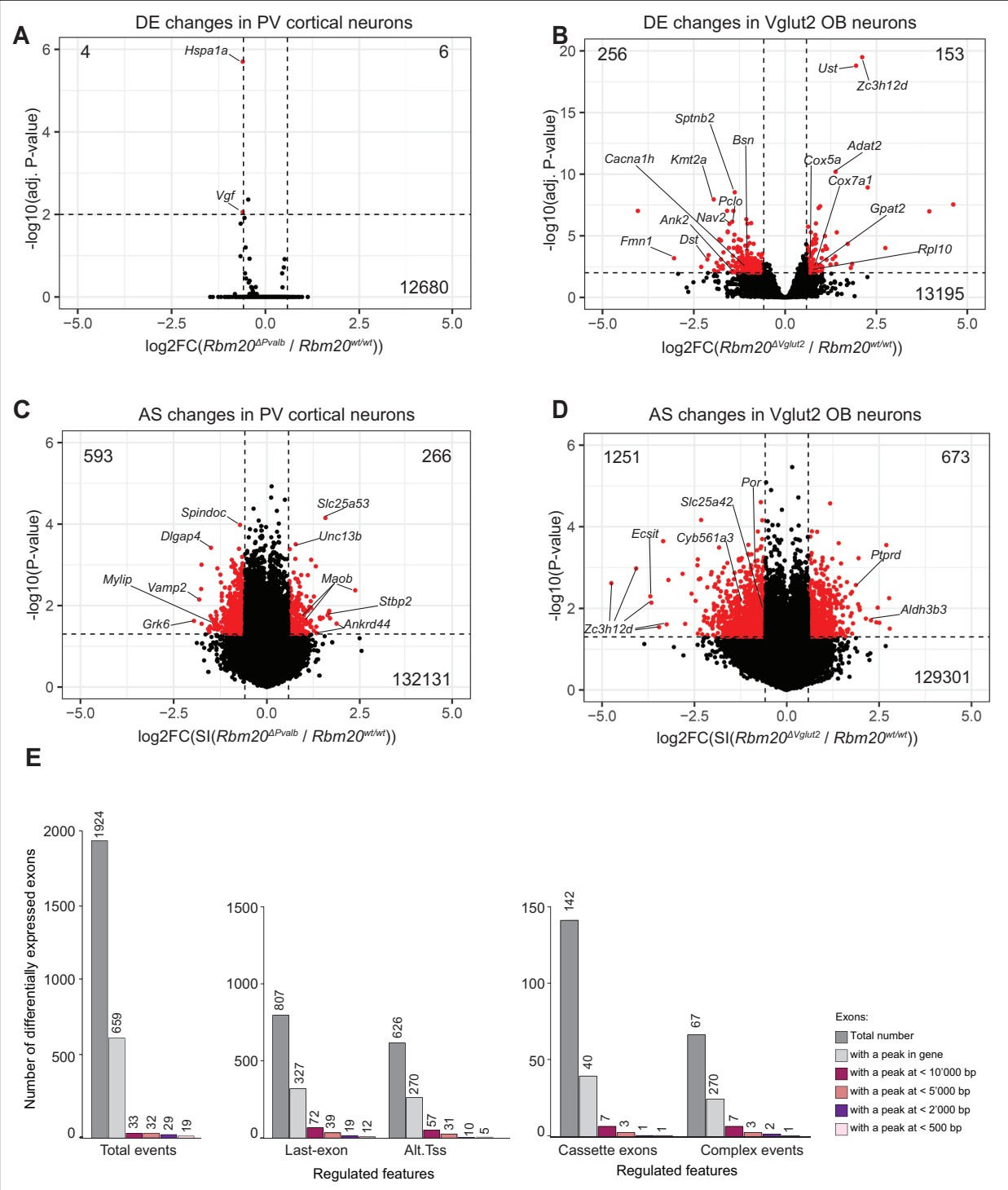

**Figure 4.** Differential gene expression and alternative exon incorporation rates in *Rbm20* conditional knockout cells. (**A**) Volcano plot of differential gene expression in RiboTrap-isolated mRNAs from *Rbm20^WT^* vs. *Rbm20ΔSlc17a6* olfactory bulb (**P35**). Significantly regulated genes shown in red, cutoff fold change (FC) of 1.5, adjusted p-value<0.01, total number of up- and downregulated noted on top. Note that *Rbm20* itself is strongly reduced, outside the axis limits, and not represented in this plot (see ***Supplementary file 4***). (**B**) Volcano plot of the differential gene expression in RiboTrap-isolated mRNAs from *Rbm20^WT^* vs. *Rbm20ΔPV* mouse neocortex (**P35**) as in panel A. The Y-chromosomal genes *Ddx3y, Uty, Kmd5d, Eif2s3y* were highly differentially expressed due to the larger number of *Rbm20* mutant males used in the Ribotrap isolations (3 wild-type females and 1 wild-type male vs. 4 knockout male mice were used for this experiment). These genes and *Rbm20* itself were excluded from the plot (see ***Supplementary file 4*** for complete data). (**C**) Volcano plot representing differentially included exons in *Rbm20ΔSlc17a6* RiboTrap-isolated mRNAs from olfactory bulb neurons. The dotted lines correspond to FC values of 1.5 and –1.5 and –log10 (p-value) of 1.3. Significantly regulated exons (FC 1.5 and p<0.05) are shown in red. (**D**) Volcano

*Figure 4 continued on next page*

*Figure 4 continued*

plot representing differentially included exons in *Rbm20ΔPV* RiboTrap-isolated mRNAs from cortical interneurons. The dotted lines correspond to FC values of 1.5 and –1.5 and –log10 (p-value) of 1.3. Significantly regulated exons (FC 1.5 and p-value<0.05) are shown in red. (**E**) Number of exons differentially expressed in *Slc17a6*[+] cells isolated from the olfactory bulb of *Rbm20ΔSlc17a6* mice and number of exons with significant RBM20 CLIP peaks in binding sites within indicated distances. Equivalent information is provided for exons divided by annotations for specific features of regulation: alternative polyadenylation (last exons), alternative transcription start sites (TSS), complex events, and cassette exons.

The online version of this article includes the following figure supplement(s) for figure 4:

**Figure supplement 1.** Normal morphological differentiation of mitral cells in the absence of RBM20.

**Figure supplement 2.** Quality control analysis of Ribo-TRAP RNA-sequencing samples.

majority of CLIP peaks were distant (>10 kb) from differentially expressed exons. This applied regardless of the type of alternative exon feature (alternative last exon, alternative transcription start sites, cassette exons, or complex alternative splicing events). Thus, we identified only one differentially regulated cassette exon with a proximal RBM20 binding event (<500 bp from the regulated exon) in the gene *Arhgef1* (*Figure 4E*). This suggests that suppression of alternative exons through proximal intronic splicing silencer elements is unlikely to be the primary essential function of neuronal RBM20.

Overall, RBM20-binding sites identified by CLIP were distributed throughout the entire length of introns and did not exhibit a particular concentration at exon-intron boundaries (*Figure 5A*). When examining the intersection of differential gene expression and CLIP data, we observed that 129 of the 256 downregulated transcripts contained CLIP peaks (50.4%), whereas only 7 of the 153 upregulated transcripts were bound by RBM20 (4.5%) (*Figure 5B*). This suggests that RBM20 loss-of-function in neuronal cells results in the loss of mRNAs, whereas the observed elevation of selective transcripts is likely to be an indirect, compensatory response. This notion is further supported by the distinct GOs of the deregulated transcripts. Downregulated genes displayed a significant enrichment in GO terms related to synaptic components and the cytoskeleton (*Figure 5C* and *Supplementary file 6*). By contrast, for the upregulated genes, there was a pronounced enrichment in mitochondrial components and ribosome composition (*Figure 5C* and see *Supplementary file 6*).

TDP-43 and MATR3, two proteins with similar domain organization to RBM20, regulate target mRNAs by suppressing aberrant splicing into cryptic exons, a phenotype most significant for genes with long introns (*Polymenidou et al., 2011*; *Ling et al., 2015*; *Attig et al., 2018*). Interestingly, intron length of transcripts showed a significant correlation with differential gene expression and the number of CLIP peaks in *Rbm20ΔSlc17a6* cells (*Figure 5D–F*). Importantly, mean intron length was substantially larger for downregulated transcripts when compared to all detected genes or to upregulated genes (*Figure 5E*). This difference was amplified when selectively examining introns of downregulated genes with CLIP peaks (*Figure 5F and G*). Finally, introns bound by RBM20 were significantly longer than expected by chance as assessed with a permutation test. Random regions with the same properties as RBM20 CLIP peak regions were generated on introns from genes expressed in our Ribo-TRAP dataset (see Materials and methods). This resulted in a mean expected intron size of 41.5 kb, which is substantially smaller than the mean size of RBM20-bound introns (59.0 kb; p-value<0.0002). Thus, long neuronal mRNAs are preferentially bound by RBM20 and particularly sensitive to loss of neuronal RBM20 protein.

## Discussion

Our work establishes an unexpected function for RBM20 in neuronal RNA metabolism. We identified RBM20 expression in two specific neuronal populations in the mouse brain. Interestingly, these two populations are derived from highly divergent lineages: glutamatergic neurons of the olfactory bulb from the rostral telencephalon and PV[+] GABAergic interneurons in the somatosensory cortex, which arise from the medial ganglionic eminence of the subcortical telencephalon. Genetic deletion of RBM20 had only a modest impact on gene expression and transcript isoforms in PV[+] interneurons but was associated with substantial alterations in glutamatergic cells of the olfactory bulb. This might be due to the higher expression of RBM20 in mitral and tufted cells as compared to PV[+] interneurons. Moreover, PV[+] interneurons express high levels of the RBM20 paralogue MATR3 (*Figure 1—figure supplement 1A*), which may have overlapping functions.

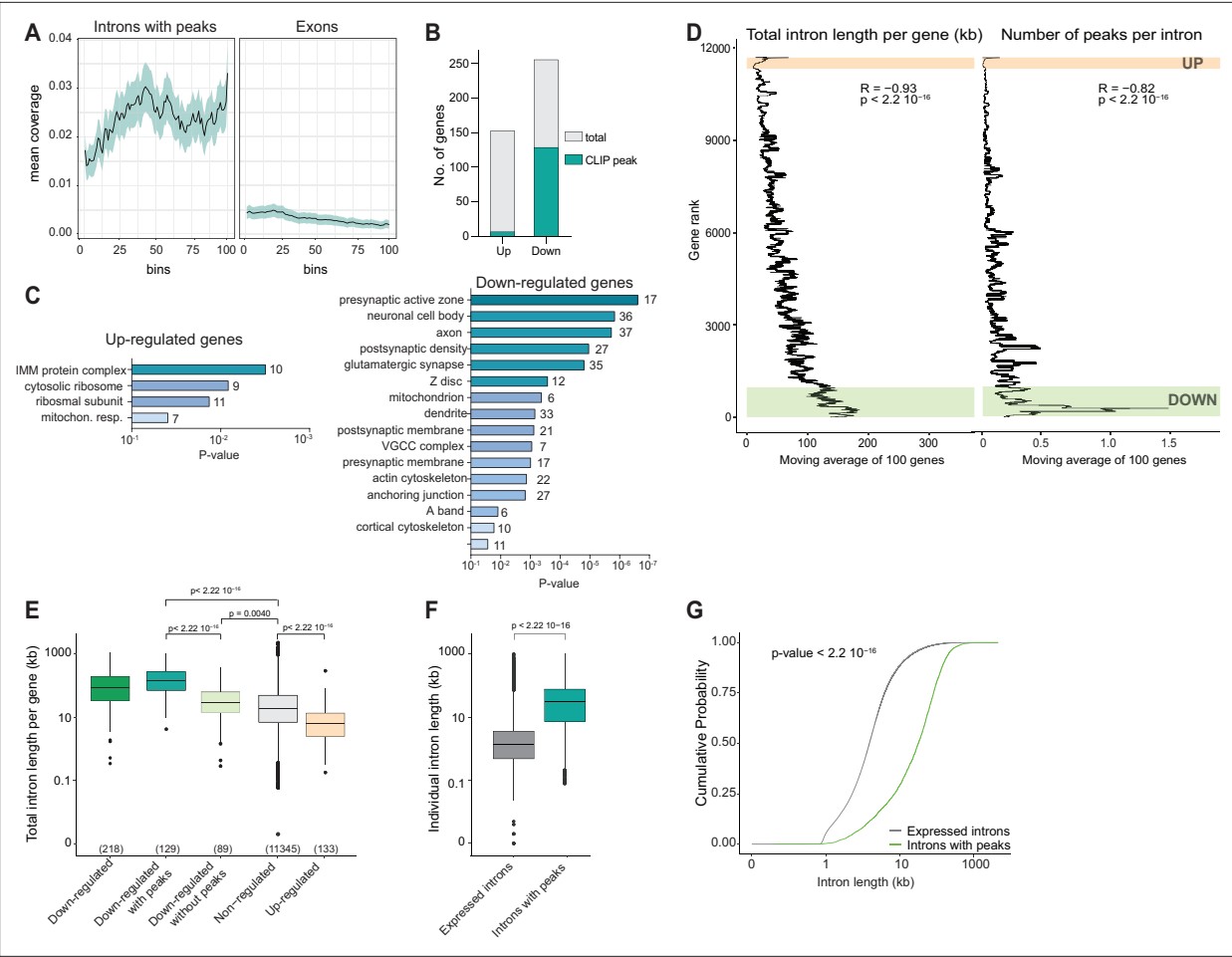

**Figure 5.** Long pre-mRNAs are depleted in *Rbm20ΔSlc17a6* mitral cells. (**A**) Metagene coverage plots of CLIP peaks across all RBM20-bound introns. Peak density across exons from the same transcripts is shown for comparison. (**B**) Total number of genes up- or downregulated (FC>1.5 and adjusted p-value<0.01) in glutamatergic cells from the olfactory bulb of *Rbm20ΔSlc17a6* mice. The fraction of genes with significant RBM20 CLIP peaks is indicated in blue. (**C**) Illustration of gene ontologies enriched amongst up- and downregulated genes (cellular component analysis with Bonferroni correction [p-value 0.05]). Minimum number of five genes identified per category. (**D**) Correlation of differential gene expression in *Rbm20ΔSlc17a6* cells, intron length, and CLIP-seq data. Genes were ranked by FC in differential gene expression and mean total intron length (left), and mean number of intronic CLIP peaks (right) for blocks of 100 genes were plotted. Ranks of the genes meeting FC cutoff for down- or upregulation are highlighted in color. Spearman's coefficients and p-values are indicated. (**E**) Boxplot showing the total intron length per gene (expressed in log$_{10}$ scale) for categories of downregulated genes (all, with or without RBM20-binding sites), nonregulated genes, and upregulated genes in *Rbm20ΔSlc17a6* in RiboTrap isolates from the olfactory bulb. Only annotated genes are plotted (number of genes shown at the bottom). p-Values from the Wilcoxon test are indicated. Medians: all downregulated genes 83.2 kb, downregulated genes with peaks: 152.4 kb; downregulated genes without peaks: ~33.6 kb; all non-regulated genes: 21.5 kb; all upregulated genes: 7.4 kb. (**F**) Boxplot (log$_{10}$ scale) illustrating the total length of introns found in genes identified in our RBM20 Ribo-TRAP dataset (gray) compared to introns presenting RBM20-binding sites (green). RBM20-bound introns exhibit a higher intron length. p-Values from the Wilcoxon test are indicated. Medians: expressed introns: 1.4 kb; introns with peaks: 30.5 kb. (**G**) Plot representing the cumulative probability distribution of intron length between the two groups of introns as in panel F. p-Value (Kolmogorov-Smirnov test) is indicated.

In cardiomyocytes, several direct targets of RBM20-dependent alternative splicing regulation contain intronic RBM20-binding sites in close proximity to regulated exons (*Maatz et al., 2014*; *van den Hoogenhof et al., 2018*). In neuronal cells, we did not observe a similar regulation of proximal exons by RBM20. Interestingly, a large fraction of transcripts downregulated in olfactory bulb RBM20 knockout cells contained intronic RBM20-binding sites. Transcripts with long introns – which are prominently expressed in the mouse brain (*Sibley et al., 2015*; *Zylka et al., 2015*) – were particularly sensitive to RBM20 loss. Thus, in neuronal cells, the splicing repressive function of RBM20 might prevent the recruitment of cryptic splice acceptor sites within long intronic segments. We hypothesize that the reduced expression of RBM20 target mRNAs observed in *Rbm20ΔSlc17a6* neurons arises from aberrant nuclear splicing, which ultimately results in the degradation of the transcripts.

The association of *RBM20* mutations with cardiomyopathies has directed substantial research efforts to uncover its function in the heart and to understand the impact of cardiomyopathy-associated mutations (*Khan et al., 2016*; *Briganti et al., 2020*; *Schneider et al., 2020*; *Fenix et al., 2021*). As for mice, human *RBM20* mRNA is also expressed in PV⁺ interneurons (*Sjöstedt et al., 2020*). There is a growing body of literature highlighting shared genetic etiology of congenital heart disease and neurodevelopmental disorders (*Homsy et al., 2015*; *Jin et al., 2017*; *Rosenthal et al., 2021*). A significant fraction of children with congenital heart disease exhibits autistic traits, including problems with theory of mind and cognitive flexibility (*Marino et al., 2012*) and the probability of an individual with congenital heart disease being diagnosed with autism was estimated to be more than twofold higher than in the typically developing population (*Gu et al., 2023*). A homozygous Ser529Arg substitution in the RNA recognition domain of RBM20 was recently identified in two brothers of consanguineous families affected by epilepsy and developmental delay (*Badshah et al., 2022*). The *RBM20* variant segregated with a second mutation in the *CNTNAP2* gene, which encodes a neuronal transmembrane protein implicated in axon-glia interactions. Thus, it remains to be explored whether alterations in neuronal RBM20 function contribute to the clinical characteristics observed in these individuals. However, given the neuronal expression and function of RBM20 identified in our study, a survey of – thus far unexplored – neurological phenotypes in *RBM20* mutation carriers might be warranted.

## Materials and methods
### Mice
All procedures involving animals were approved by and performed in accordance with the guidelines of the Kantonales Veterinärat Basel-Stadt. Male and female mice were used in this study. All other mice were in C57BL/6J background. *Rpl22*^HA (RiboTag) mice (*Sanz et al., 2009*), *CamK2*^Cre (*Tsien et al., 1996*), *SST*^Cre (*Taniguchi et al., 2011*), *Pvalb*^Cre mice (*Hippenmeyer et al., 2005*), *Rosa26*^LSL-tdTomato reporter mice (*Madisen et al., 2010*), and *Slc17a6*^Cre mice (*Vong et al., 2011*) were obtained from Jackson Laboratories (Jax stock no: 011029, 017320, and 007909, 028863, 007914 tdTomato, and 016963). *Rbm20* ^floxed mice and *Rbm20* constitutive KO mice (*Khan et al., 2016*) were backcrossed for at least four generations to a C57BL/6J background. *Tbx21*^Cre mice (*Haddad et al., 2013*) were kindly provided by Dr. R. Datta's Laboratory.

The *Rbm20* knock-in mouse line was generated using Crispr-Cas9 in collaboration with the Center for Transgenic Models (CTM) at the University of Basel. gRNAs targeting the last intron of *Rbm20* and template 'coin allele' construct were injected together with RNA encoding for Crispr-Cas9 nuclease into C57BL/6J zygotes. The surviving embryos were transferred into recipient females. The *coin* module fused to a histidine-biotin-histidine-3xHA tag. The *coin allele* is inserted in an orientation opposite to the gene's direction of transcription. The gRNA used was: 5' TTGAGTCGGGGGTCCC ACTG 3'. The 1311 bp single-stranded megamer containing the upstream homology sequence (lower case), *coin module* (upper case letters), and downstream homology sequence (lower case) and the optimized codon usage (lower case) used:

5'ggcgaggctgctgctggagagccctgatttcttctctgtttgactcgcgaattctgaggggataagcgccctgcatatgtatgc
attcttctttgggagcctgcagccaccttcatgcccagtaaggctatgcttactgtgccagatcaccccctgtaggctcacatagagccatg
accagcaacagcatagcgggatttccagaggcttcactgaggcagctatgacctgctcttgcctcccagggcatCCCCAGTACC
GTTCGTATAatgtatgcTATACGAAGTTATGGGCCCCTCTGCTAACCATGTTCATGCCTTCTTCTTTTTCCT
ACAGAAGTACCTGTCTCAGCTGGCAGAGGAGGgactcAAGGAGACGGAGGGGACAGACAGCCCA
AGCCCCGAGCGTGGTGGGATTGGTCCACACTTGGAAAGGAAGAAGCTAGCtGGcCAcCATCACC
ACCAcCATGGTGCcGCTGGAAAGGCCGGTGAAGGTGAAATCCCTGCCCCTCTTGCTGGTACaGT
TTCTAAGATACTcGTAAAAGAAGGTGACACTGTTAAAGCTGGTCAAACAGTTCTGGTGCTGGAG
GCcATGAAAATGGAGACAGAAATTAACGCTCCTACTGACGGAAAAGTTGAAAAGGTGTTAGTTA
AGGAAAGAGATGCTGTTCAAGGTGGTCAAGGTCTAATCAAGATCGGCGTTGCAGGTCATCAcCA
CCAtCATCAcGGcGCCgccgggTATCCCTACGATGTGCCTGACTATGCTgctggcTATCCTTACGACG
TGCCCGATTATGCAgccggcTATCCATACGATGTCCCAGATTACGCTgccTAGGATCTTTTTCCCTCT
GCCAAAAATTATGGGGACATCATGAAGCCCCTTGAGCATCTGACTTCTGGCTAATAAAGGAAAT
TTATTTTCATTGCAATAGTGTGTTGGAATTTTTTGTGTCTCTCACTCGGAAGGACATATGGGAGGGCA
AATCATTTAAAACATCAGAATGAGTATACCGTTCGTATAgcatacatTATACGAAGTTATTGGGACCC
CCGACTCAAggtctcctgatgaatgctaactttctaagttgcctgacttgagtcagctggcacctgccctgtgggtcagacttcttca

cttttcacacttgtggtttggagtaaagtgggagaggctgtagagactgaggcattcattctgccaaggcccctgacagaaacgctac
ctgagatggctgtggcagaggctcctggctccctgataaaaggtgtaccagggaaacgtgagctgaggtgggagggagtgagg'.
The entire insert sequence is highlighted in capital letters.

Activation by Cre-recombinase inverts the *coin* module, resulting in alternative splicing of the tagged exon. We observed only very low rates of Cre-mediated inversion upon crossing to Cre-driver mouse lines. Thus, a germline inverted allele was generated and used in all experiments. This constitutively tagged *Rbm20HA* mouse strain was deposited at EMMA.

All mouse lines were maintained on a C57Bl6/J background. Both males and females were used for all the experiments unless stated otherwise in the respective Materials and methods sections.

## Surgeries and stereotaxic injections

Recombinant adeno-associated viruses with the rAAV2 capsid (*Tervo et al., 2016*) were produced in HEK293T cells. In brief, initially, a DNA mixture consisting of 70 µg AAV helper plasmid, 70 µg AAV vector, and 200 µg pHGTI-adeno1 is prepared in a Falcon tube and added at a 1:4 DNA:PEI ratio to each cell plate. After 48–60 hr, cells are collected and debris is collected by centrifugation at 4000 rpm, 4°C for 20 min. The supernatant containing the virus is subsequently purified through Opti-Prep Density Gradient Medium (Sigma, Cat. No D1556). Viral preparations are concentrated in 100 K Millipore Amicon columns at 4°C. The virus samples were then suspended in PBS 1×, aliquoted, and stored at –80°C. Viral titers were determined by qPCR and were >$10^{12}$ particles/ml.

Mice (postnatal days 24–27) were placed on a heating pad in a stereotaxic frame (Kopf Instrument) under isoflurane anesthesia. A small incision (0.5–1 cm) in the skin overlying the area of interest was made, and bilateral viral injections were performed in the posterior piriform cortex using a Pico-spritzer III pressure injection system (Parker) with borosilicate glass capillaries (length 100 mm, OD 1 mm, ID 0.25 mm, wall thickness 0.375 mm). Coordinates: ML = +2.2 mm, AP = +2.35 mm, DV = –3.95 mm from Bregma. A volume of 100 nl of virus was delivered to each side, through repeated puffs over several minutes. Viruses used were rAAV2-CAG-DiO-eGFP or rAAV2-SYN-Cre virus, driving Cre-dependent eGFP expression from the chicken beta actin promoter and human synapsin promoter, respectively. Ten days after viral infection, mice were anesthetized and transcardially perfused. Position of the viral injection site was confirmed on coronal brain slices using DAPI-stained sections on an AxioScan.Z1 Slide scanner (Zeiss) using a 20× objective.

## Immunochemistry and imaging

Animals (males and females) from postnatal days 25–40 were anesthetized with ketamine/xylazine (100/10 mg/kg i.p.) and transcardially perfused with fixative (4% paraformaldehyde). The brains or hearts were postfixed overnight in the same fixative at 4°C and washed three times with 100 mM phosphate buffer. Coronal brain slices were cut at 40 µm with a vibratome (Leica Microsystems VT1000).

For immunohistochemistry, brain sections were processed as previously described (*Traunmüller et al., 2023*). In brief, brain slices were kept for 1.5 hr with a PBS-based blocking solution containing 0.1% Triton X-100 and 5% normal donkey serum (NDS) and subsequently incubated with primary antibodies in blocking solution at 4°C overnight. Secondary antibodies were diluted in 5% NDS in PBS containing 0.05% Triton X-100 to a final concentration of 0.5 or 1.0 µg/ml and incubated with sections for 2 hr at room temperature.

Image stacks were acquired at room temperature on a laser-scanning confocal microscope (Zeiss LSM700) using a 40× Apochromat objective with numerical aperture 1.3, controlled by Zen 2010 software. Following acquisition, images were processed and assembled using Fiji (*Schindelin et al., 2012*), OMERO, and Adobe Illustrator software.

For quantifications of RBM20-positive neurons in the olfactory bulb, tile-scan images from 30 µm slices from the olfactory bulb of P35 mice were acquired. Mean intensity analyses of RBM20 signal were performed using Fiji (*Schindelin et al., 2012*) using a custom-made Python script, as previously described (DOI–https://github.com/imcf-shareables/3D_spots_count/blob/main/README.md). In brief, neuronal cells were identified based on the nuclear DAPI signal. The mean intensity of RBM20 protein in each nucleus was then measured, and the background signal was subtracted.

For the characterization of RBM20 subnuclear localization, brain and heart samples from *Rbm20* WT and *Rbm20* cKO mice (P35–40) were anesthetized with ketamine/xylazine (100/10 mg/kg i.p.) and transcardially perfused with fixative (4% paraformaldehyde). The brains and hearts were postfixed

overnight in the same fixative at 4°C and washed three times with 100 mM phosphate buffer. Coronal brain slices were cut at 40 μm with a vibratome (Leica Microsystems VT1000). Brain samples were immersed in 15% and subsequently in 30% sucrose in 1× PBS for 48 hr, cryoprotected with Tissue-Tek Optimum Cutting Temperature, and frozen at –80°C until use. Tissue was sectioned at 40 μm on a cryostat (Microm HM560, Thermo Scientific) and collected in 1× PBS. Immunohistochemistry and imaging were performed as previously described.

## Antibodies

The following commercially available antibodies were used: rat-anti-HA (Roche, #11867431001) and rabbit-anti-GAPDH (Cell Signaling #5174), rabbit-anti-MAP2 (Synaptic Systems #188002), rabbit-anti-RFP (Rockland #600-401-379), chicken-anti-GFP (Aves Labs Inc #GFP-1020), goat anti-Parvalbumin (Swant, PVG213), rabbit-anti-NeuN (Novus Biologicals, Cat. No. NBP1-77686SS). Secondary antibodies coupled to horseradish peroxidase (HRP) or fluorescent dyes were from Jackson ImmunoResearch (goat anti-rabbit HRP #111-035-003; goat anti-rat HRP #112-035-143), donkey anti-rat IgG-Cy3 and Cy5 (Jackson ImmunoResearch, 712-165-153, 706-175-148) donkey anti-goat IgG-Cy3, and donkey anti-chicken IgG-Cy3 (Jackson ImmunoResearch, 705-165-147, 703-165-155).

For the generation of polyclonal anti-RBM20 antibodies, a synthetic peptide consisting of the RBM20 C-terminus was used: C+PERGGIGPHLERKKL (N- to C-terminus, C+ indicates a cysteine added to the N-terminus for thiol-mediated coupling). The synthetic peptide was conjugated to keyhole limpet hemocyanin for immunization of rabbits and guinea pigs (Eurogentec, Belgium). Resulting sera were affinity-purified on the peptide antigen and the specificity of the resulting antibodies was confirmed using lysates and tissue sections from *Rbm20* knockout mice.

## Tissue clearing and anatomical reconstruction of mitral cells

The olfactory bulb and part of the anterior prefrontal cortex of rAAV2-infected mice were dissected after transcardial perfusion and cleared using the Cubic L protocol (*Tainaka et al., 2018*). Olfactory bulbs were placed in a 5 ml Eppendorf tube filled with pre-warmed CUBIC L solution (10% N-butyl-diethanolamine and 10% Triton X-100 dissolved in MilliQ water). The tissue was incubated on a shaking plate at 37°C for 48 hr. Cleared bulbs were washed in 50 mM PBS three times for 10 min and then cut into two coronal halves under a Binocular Stereo Microscope (Olympus #MVX10). The two halves of each bulb (anterior and posterior) were then embedded in 1% agarose in TBE solution in a glass-bottom imaging chamber (Ibidi, Cat. No. 80426). Z-stacks of GFP+ neurons with the soma residing in the MCL of the olfactory bulb were acquired on a Olympus two-photon microscope fitted with a MaiTai eHP laser (Spectra-Physics) and a 25× objective with 1.05 NA. A volume of up to 2×2 mm$^2$ (xy) × 500–700 μm in depth was acquired in tiles up to 4×4, with x=0.995 μm, y=0.995 μm, and z=3 μm pixel size. Laser power was linearly adjusted with imaging depth and typically ranged between 0.5 and 20 mW. GFP+ mitral cells in the two-photon z-stacks were traced neurons using Neurolucida 360 software.

For the two-photon image analysis, a total of 10 neurons (5 neurons per genotype from at least 3 biological replicates) were analyzed. Both apical and lateral dendrites of mitral cells were traced semiautomatically by using the user-guided 3D image detection algorithm. Tracings were checked and corrected manually when needed. Subcellular components, such as spines and other small protrusions, were not traced. The following parameters were extracted from the software Neurolucida Explorer for each traced neuron: the number of dendrites from different centrifugal orders (i.e. primary dendrites, secondary dendrites, etc.) and the total dendritic length of the glomeruli tufts. Average values were calculated for each neuron analyzed. Graphs and statistical analyses (t-tests) were made using GraphPad Prism.

## Fluorescent in situ hybridization

FISH was performed using the RNAscope Fluorescent Multiplex Kit (Advanced Cell Diagnostics, Cat. No. 320851). P25 mouse brains were snap-frozen in liquid nitrogen, and 15 μm coronal sections were cut on a cryostat (Microm HM560, Thermo Scientific). Sections were fixed at 4°C overnight with 4% paraformaldehyde in 100 mM PBS, pH 7.4.

Images were acquired at room temperature with an upright LSM700 confocal microscope (Zeiss) using 40× Apochromat objectives (NA = 1.3). Stacks of 10–15 μm thickness (0.44 μm interval between

image planes) were acquired from layer 5 (**L5**) of the primary somatosensory area (**S1**). Genetically marked cell types were identified based on the presence of transcripts encoding the *tdTomato* marker. Commercially available probes were used to detect *Rbm20* (ACD #549251) and *tdTomato* (ACD #317041). A region of interest (ROI) was drawn to define the area of the cell and dots in the ROI were manually counted throughout the image z-stacks. The number of dots in the ROI was then normalized to the cell area (measured in μm [*Attig et al., 2018*]). Images were assembled using Fiji and Adobe Illustrator software.

For quantification of *Slc17a6, Tbr2,* and *Rbm20* transcripts expression in mitral and tufted neurons of the olfactory bulb, P25 animals were euthanized and the brains were harvested and processed as described above. Stacks of 10–15 μm width (0.44 μm interval between stacks) were acquired from olfactory bulb slices at room temperature with an upright LSM700 confocal microscope (Zeiss) using 40× Apochromat objectives. An ROI was drawn to define the area of each cell residing either in the MCL or GL of the olfactory bulb. Dots in the ROIs were detected automatically throughout the z-stacks for each channel, using a custom-made Python script, as described in (DOIhttps://github.com/imcf-shareables/3D_spots_count/blob/main/README.md). The following commercial probes were used: *Rbm20* (549251), *slc17a6* (319171), *Tbr2* (Eomes): (429641). Images from three mice were used for the quantification (two images per slice). *Gad2* (415071) in situ hybridization was performed on 15 μm olfactory bulb slices of P25 mice.

## Biochemical procedures

Mouse tissues were extracted on ice and lysed in 50 mM Tris-HCl pH 8.0, 150 mM NaCl, 0.1% SDS, 5 mM EDTA, 1% Igepal, and protease inhibitor (Roche Complete). The lysate was sonicated on ice (100 Hz amplitude 0.5 cycles × 10 pulses) and centrifuged for 20 min at 13,000×*g* at 4°C. Proteins in supernatant were analyzed by gel electrophoresis on 4–20% gradient polyacrylamide gels (Bio-Rad, 4561093) and transferred onto nitrocellulose membrane (Bio-Rad, 1704158). Membranes were blocked with 5% non-fat dry milk (NFDM, PanReac AppliChem, Cat. No. A0830) blocking buffer in TBS-T 1× for 2 hr at room temperature, and protein detection was by chemiluminescence with HRP-conjugated secondary antibodies (WesternBright Quantum, Advasta, Cat. No. K-12043 D20, K-12042 D20).

RiboTRAP purification (*Heiman et al., 2008*; *Sanz et al., 2009*) was performed with some modifications as described in *Di Bartolomei and Scheiffele, 2022*.

## Targeted LC-MS sample preparation and analysis

Murine nuclear extracts of heart tissue were lysed in 100 mM triethylammonium bicarbonate pH 8.5/5% SDS/10 mM tris(2-carboxyethyl)phosphin using 20 cycles of sonication (30 s on/30 s off per cycle) on a Bioruptor system (Dianode) followed by heating to 95°C for 10 min. Protein extracts were alkylated using 15 mM iodoacetamide at 25°C in the dark for 30 min. For each sample, 50 μg of protein lysate was captured, digested, and desalted using STRAP cartridges (Protifi, NY, USA) following the manufacturer's instructions. Samples were dried under vacuum and stored at –80°C until further use.

For parallel reaction-monitoring assays (*Peterson et al., 2012*), three proteotypic peptides derived from RBM20 were selected for assay development (ASPPTESDLQSQACR, QGFGCSCR, and SGSPG-PLHSVSGYK). A mixture containing 100 fmol of each heavy reference peptide (JPT, Berlin, Germany) including iRT peptides (Biognosys, Schlieren, Switzerland) was used. The setup of the μRPLC-MS system was as described previously (*Ahrné et al., 2016*). Peptides were analyzed per LC-MS/MS run using a linear gradient ranging from 95% solvent A (0.15% formic acid, 2% acetonitrile) and 5% solvent B (98% acetonitrile, 2% water, 0.15% formic acid) to 45% solvent B over 60 min at a flow rate of 200 nl/min. Mass spectrometry analysis was performed on a Q-Exactive HF mass spectrometer equipped with a nanoelectrospray ion source (Thermo Fisher Scientific) as described previously (*Hauser et al., 2022*). The acquired raw files were database-searched against a *Mus musculus* database (Uniprot, download date: 2020/03/21, total of 44,786 entries) by the MaxQuant software (version 1.0.13.13). To control for variation in sample amounts, the total ion chromatogram (only comprising peptide ions with two or more charges) of each sample was determined by Progenesis QI (version 2.0, Waters) and used for normalization. The datasets of this study are deposited on MassIVE (code: MSV000093344) and PRIDE (code: PXD046806).

## Library preparation and deep sequencing

For all the RNA-sequencing experiments, the quality of RNA integrity was analyzed using an RNA 6000 Pico Chip (Agilent, 5067-1513) on a Bioanalyzer instrument (Agilent Technologies) and only RNA with an integrity number higher than 7 was used for further analysis. RNA concentration was determined by fluorometry using the QuantiFluor RNA System (Promega #E3310) and 50 ng of RNA was reverse-transcribed for analysis of marker enrichment by quantitative PCR (see extended methods).

Up to five biological replicates per neuronal population and genotype were analyzed. Library preparation was performed, starting from 50 ng total RNA, using the TruSeq Stranded mRNA Library Kit (Cat. No. 20020595, Illumina, San Diego, CA, USA) and the TruSeq RNA UD Indexes (Cat. No. 20022371, Illumina, San Diego, CA, USA). 15 cycles of PCR were performed. Libraries were quality-checked on the Fragment Analyzer (Advanced Analytical, Ames, IA, USA) using the Standard Sensitivity NGS Fragment Analysis Kit (Cat. No. DNF-473, Advanced Analytical).

The samples were pooled to equal molarity. The pool was quantified by fluorometry using the QuantiFluor ONE dsDNA System (Cat. No. E4871, Promega, Madison, WI, USA) before sequencing. Libraries were sequenced in paired-end mode (151 bp reads) (in addition: 8 bases for index 1 and 8 bases for index 2) setup using the NovaSeq 6000 instrument (Illumina) and the S1 flow cell loaded at a final concentration of 340 pM with 1% PhiX.

Primary data analysis was performed with the Illumina RTA version 3.4.4. On average per sample: 49±4 millions pass-filter reads were collected on this flow cell.

## Quality control of ribotag pull-downs

The enrichment and de-enrichment of markers following neuronal markers were tested: for the olfactory bulb pull-down samples: *Rbm20*, *Slc17a6*, *Slc17a7*, *Tbr2*, *Pcdh21*, Slc32a1, *Gfap*, *Gad67*. For pull-downs from the cortex of PVCre mice: *Rbm20*, *Pvalb*, *Slc32a1*, *Gad67*, *Slc17a7*, *Gfap*. In both cases, *Gapdh* mRNA was used as a housekeeping gene for normalization. The fold enrichment and de-enrichment values of each marker were calculated for each cell population in immunoprecipitated RNA, comparing it to input purifications. Only samples that showed correct enrichment or de-enrichment for excitatory or inhibitory neuronal markers and a de-enrichment for glia markers were further used for sequencing. DNA oligonucleotides were used with FastStart Universal SYBR Green Master (Roche, 4913914001) and comparative $C_T$ method. For each assay, three technical replicates were performed and the mean was calculated. RT-qPCR assays were analyzed with the StepOne software. DNA oligonucleotides used (name and sequence 5'–3' are indicated):

List of primer sequences:

| Primer name | Sequence |
|---|---|
| Rbm20 | Forward (Sense) TGCATGCCCAGAAATGCCTGCT Reverse (AntiSense) AAAGGCCCTCGTTGGAATGGCT |
| Tbr2 | Forward (Sense) ATAAACGGACTCAACCCCACC Reverse (AntiSense) CCCTGCATGTTATTGTCCGC |
| Pchd21 | Forward (Sense) ATCACTGTCAACGACTCAGACC Reverse (AntiSense) GTCAATGGCAGCTGAGTTTTCC |
| Slc17a6 | Forward (Sense) GCATGGTCTGGTACATGTTCTG Reverse (AntiSense) GACGGGCATGGATGTGAAAAAC |
| Gad67 | Forward (Sense) GTACTTCCCAGAAGTGAAAC Reverse (AntiSense) GAATAGTGACTGTGTTCTAGG |
| Gfap | Forward (Sense) CTCGTGTGGATTTGGAGAG Reverse (AntiSense) AGTTCTCGAACTTCCTCCT |
| Slc17a7 | Forward (Sense) ACCCTGTTACGAAGTTTAACAC Reverse (AntiSense) CAGGTAGAAGGTCCAGCTG |
| Slc32a1 | Forward (Sense) CGTGACAAATGCCATTCAG Reverse (AntiSense) AAGATGATGAGGAACAACCC |
| Pvalb | Forward (Sense) CATTGAGGAGGATGAGCTG Reverse (AntiSense) AGTGGAGAATTCTTCAACCC |

## RNA-sequencing data analysis

Initial gene expression and alternative splicing analysis were done in collaboration with Geno-Splice Technology, Paris, as previously described (*Furlanis et al., 2019*). In brief, sequencing, data quality, reads repartition (e.g. for potential ribosomal contamination), and insert size estimation were performed using FastQC, Picard-Tools, Samtools, and RSeQC tool packages. Reads were mapped using STAR (version 2.4.0) (*Dobin et al., 2013*) on the mm10 Mouse genome assembly. The input read count matrix was the same as used for the splicing analysis. Two samples from the olfactory bulb were excluded from the analysis after the quality control of the data.

Read counts were summarized using featureCounts (*Liao et al., 2014*). For each gene present in the FASTDB v2021_4 annotations, reads aligning on constitutive regions (that are not prone to alternative splicing) were counted. Based on these read counts, normalization and differential gene expression were performed using DESeq2 (values were normalized to the total number of mapped reads of all samples) (*Love et al., 2014*) on R (version 3.5.3). Genes were considered as expressed if their FPKM value is greater than 96% the background FPKM value based on intergenic regions. A gene is considered expressed in the comparison if it is expressed in at least 50% of samples in at least one of the two groups compared. Results were considered statistically significant for adjusted p-values≤0.01 (Benjamini-Hochberg for p-value adjustment as implemented in DESeq2) and $\log_2$(FC)≥1.5 or or ≤–1.5. For the principal component analysis, counts were normalized using the variance stabilizing transform as implemented in DESeq2. The internal normalization factors of DESeq2 were used to normalize the counts for generation of heatmaps. The alternative splicing analysis was performed by calculating a splicing index (SI), which is the ratio of read density on the exon of interest and the read density on constitutive exons of the same gene. The $\log_2$ fold change (FC) and p-value (unpaired Student's t-test) were calculated by pairwise comparisons of the respective SI values. Results were considered significantly different for p-values≤0.01 and $\log_2$(FC)≥1 or ≤–1.

For the generation of volcano plots, exons and genes with *NA* or *Inf* values were removed to prevent bias caused by genes or exons with very low expression. Plots were created in R with the *ggplot2* package.

## CLIP and library preparation

The CLIP experiments were performed according to the seCLIP protocol (*Van Nostrand et al., 2017a*) with some minor modifications (*Traunmüller et al., 2023*). Olfactory bulbs from seven mice were pooled for each biological replicate, and for heart tissue, one heart was used per biological replicate. Samples were flash-frozen and ground on dry ice first in a metal grinder and a porcelain mortar. The frozen powder was transferred into a plastic Petri dish and distributed in a thin layer. The samples were UV-cross-linked three times at 400 mJ/cm² on dry ice with a UV-cross-linker (Cleaver Scientific). The powder was mixed and redistributed on the Petri dish before each UV exposure. After cross-linking, the powder was collected in 3.5 ml (olfactory bulbs) or 5.5 ml (heart) in lysis buffer (50 mM Tris-HCl pH 7.5, 100 mM NaCl, 1% NP-40, 0.1% SDS, 0.5% sodium deoxycholate, complete protease inhibitors, Roche) and 4 U/ml Turbo-DNase (Thermo Fisher). Samples were further processed as described in detail below.

For the clip samples preparation, the lysate was transferred into a glass homogenizer and homogenized by 30 strokes on ice. 1 ml aliquots of homogenized tissue were transferred to 2 ml tubes, 10 µl of RNaseI (Thermo Fisher) diluted in PBS (1:5–1:40) were added to each tube. Samples were incubated at 37°C with shaking for 5 min at 1200× rpm and then put on ice. 10 µl RNasin RNase inhibitor (40 U/µl, Promega) was added to each tube. Samples were mixed and centrifuged at 16,000 × *g* for 15 min at 4°C. The supernatants were transferred to a new tube, and 60 µl from each sample was taken and further processed for sized-matched INPUT (SMIn). 10 µl HA-magnetic beads (Pierce) was added to each sample and incubated at 4°C for 4 hr in a rotating shaker. Following incubation, the beads were washed 2× with a high salt wash buffer (50 mM Tris-HCl pH 7.5, 1 M NaCl, 1 mM EDTA, 1% NP-40, 0.1% SDS, 0.5% sodium deoxycholate), 2× with the lysis buffer, 2× with low salt wash buffer (20 mM Tris-HCl pH 7.5, 10 mM MgCl₂, 0.2% Tween-20) and 1× with PNK buffer (70 mM Tris-HCl pH 6.5, 10 mM MgCl₂). Beads were resuspended in 100 µl PNK mix (70 mM Tris-HCl pH 6.5, 10 mM MgCl₂, 1 mM DTT, 100 U RNasin, 1 U TurboDNase, 25 U Polynucleotide-Kinase [NEB]) and incubated at 37°C for 20 min on a shaking termomixer (1200× rpm). Upon RNA dephosphorylation, the beads were washed (2× high salt, 2× lysis, and 2× low salt buffers as before) and additionally with

1× Ligase buffer (50 mM Tris-HCl pH 7.5, 10 mM MgCl$_2$). Beads were then resuspended in 50 μl ligase mix (50 mM Tris-HCl pH 7.5, 10 mM MgCl$_2$, 1 mM ATP, 3% DMSO, 15% PEG8000, 30 U RNasin, 75 U T4 RNA-ligase [NEB]). 10 μl of the beads/ligase mix was transferred to a new tube, and 1 μl of pCp-Biotin (Jena Bioscience) was added to validate IP of the RNA-protein complexes by western blot. 4 μl of the RNA-adaptor mix containing 40 μM of each InvRiL19 & InvRand3Tr3 (IDT) was added to the remaining of the samples (40 μl). Samples were incubated at room temperature for 2 hr for adaptor ligation. Samples were washed 2× with high salt, 2× with lysis, and 1× with low salt buffers. Finally, beads were resuspended in 1× LDS sample buffer (Thermo Fisher) supplemented with 10 μM DTT and incubated for 10 min at 65°C, shaking on a thermomixer at 1200× rpm. Eluates or inputs were loaded on 4–12% Bis-Tris, 1.5 mm gel (Thermo Fisher) and separated at 130 V for ~1.5 hr. Proteins were transferred overnight at 30 V to a nitrocellulose membrane (Amersham). The membranes were placed in a 15 cm Petri dish on ice, and an area between 55 and 145 kDa was cut out into small pieces and transferred into a 2 ml tube.

RNA extraction, reverse transcription using InvAR17 primer, cDNA clean-up using silane beads (Thermo Fisher), second adaptor ligation (InvRand3Tr3), and cDNA purification steps were performed as previously described (*Van Nostrand et al., 2016*). The sequencing libraries were amplified using Q5-DNA polymerase (NEB) and i50X/i70X Illumina indexing primers (IDT). Final libraries were amplified with 14 cycles. Libraries were purified and concentrated with ProNEX size-selective purification system (Promega) using sample/beads ratio of 1/2.4. Samples were loaded on a 2% agarose gel, and the area corresponding to the size between 175 and 350 bp was cut out. The amplified and purified libraries were then extracted from the gel using a gel extraction kit (Machery&Nagel) and eluted with 16 μl.

The concentrations and the size distributions of the libraries were determined on the Fragment Analyzer system (Agilent). 75 bp single-end sequencing was performed on the NextSeq500 platform using Mid Output Kit v2.5 (75 cycles).

Adaptor and primer sequences used in this study:

| Name | Sequence |
|---|---|
| InvRi L19 | /5Phos/rArGrArUrCrGrGrArArGrArGrCrArCrA rCrGrUrC/3SpC3/ |
| InvRand3Tr3 | /5Phos/NNNNNNNNNNAGATCGGAAGA GCGTCGTGT/3SpC3/ |
| InvA R17 | CAGACGTGTGCTCTTCCGA |
| i501 | AATGATACGGCGACCACCGAGATCTACACTATAGCCTA CACTCTTTCCCTACACGACGCTCTTCCGATC*T |
| i502 | AATGATACGGCGACCACCGAGATCTACACATAGAGGC ACACTCTTTCCCTACACGACGCTCTTCCGATC*T |
| i503 | AATGATACGGCGACCACCGAGATCTACACCCTATCCT ACACTCTTTCCCTACACGACGCTCTTCCGATC*T |
| i504 | AATGATACGGCGACCACCGAGATCTACACGGCTCTGAA CACTCTTTCCCTACACGACGCTCTTCCGATC*T |
| i701 | CAAGCAGAAGACGGCATACGAGATCGAGTAATGTGACT GGAGTTCAGACGTGTGCTCTTCCGATC*T |
| i702 | CAAGCAGAAGACGGCATACGAGATTCTCCGGAGTGACT GGAGTTCAGACGTGTGCTCTTCCGATC*T |
| i703 | CAAGCAGAAGACGGCATACGAGATAATGAGCGGTGACT GGAGTTCAGACGTGTGCTCTTCCGATC*T |
| i704 | CAAGCAGAAGACGGCATACGAGATGGAATCTCGTGACT GGAGTTCAGACGTGTGCTCTTCCGATC*T |

X* = Phosphorthioated base

seCLIP data processing was performed as described (*Van Nostrand et al., 2016*; *Van Nostrand et al., 2017b*; *Van Nostrand et al., 2020*). In brief, raw reads were processed to obtain unique CLIP tags mapped to mm10 using Clipper (https://github.com/YeoLab/clipper ; *Lovci et al., 2025*; https://github.com/YeoLab/eclip ; *Yee and Domissy, 2022*). Reads from replicates 1 and 2 from the olfactory bulb were concatenated. Peak normalization was performed by processing the SMInput samples using

the same peak calling pipeline. Irreproducible discovery rate (IDR) analysis was performed to identify reproducible peaks across biological replicates (*Li et al., 2011*). IDR (https://github.com/nboley/idr; *Boley, 2017*) was used to rank seCLIP peaks by the fold change over the size-matched input. Clip peaks were called based on IDR<0.05. We observed some short highly represented sequences that were not specific to RBM20 seCLIP isolations, which were excluded based on peak shape and width (<30 bp) using StoatyDive (*Heyl and Backofen, 2021*).

For motif discovery, cross-link-induced truncation sites (CITS) were called using the CTK pipeline (*Shah et al., 2017*). Briefly, unique tags from replicates were combined, and CITS were called by requiring FDR<0.001. Sequences from –10 bp to +10 bp from called CITS were used as input sequences for DREME software (*Bailey et al., 2009*; *Bailey et al., 2015*; *Nystrom and McKay, 2021*). As a control, sequences of the same length located 500 bp upstream of the CITS site (–510 to –490 bases) were used. Enrichment of the UCUU motif at the CITS sites was calculated.

## Analysis of RBM20-bound intron length

For investigating the intron length of RBM20-bound introns, we performed a permutation test to calculate an empirical p-value. We generated 5000 sets of random genomic coordinates, mirroring the length distribution and quantity of RBM20 seCLIP peaks. These coordinates were confined to the intronic regions of genes identified in the *Slc17a6* RiboTrap datasets.

In each permutation, we computed the mean length of all introns that included these random regions. The resulting distribution, based on 6000 mean intron lengths, yielded a mean of ~41.4 kb nucleotides and a standard deviation of ~288. The average length of introns containing RBM20 seCLIP peaks was ~59.0 kb, corresponding to a z-score of 60.95 and yielding a p-value of $1.666667 * 10^{-4}$. This p-value is constrained by the number of permutations conducted.

## GO analysis

All the GO analyses were performed by using a statistical overrepresentation test and the cellular component function in PANTHER (https://pantherdb.org/). All genes being detected as expressed in the Ribo-TRAP RNA-sequencing data were used as reference. GO cellular component annotation dataset was used, and Fisher's exact test and Bonferroni correction for multiple testing were applied. GO terms with at least five genes and with p-value<0.05 were considered as significantly enriched. Significant GO terms were plotted in Prism 9.

For seCLIP GO analysis, any gene that had significant peak expression in the CLIP dataset either for olfactory bulb or heart samples was used, and all genes being detected as expressed in the seCLIP size-matched input samples were used as reference.

## Statistical methods and data availability

Sample sizes were determined based on the 3R principle, past experience with the experiments and literature surveys. Pre-established exclusion criteria were defined to ensure success and reliability of the experiments: for stereotaxic injection, all mice with mis-targeted injections were excluded from analysis (e.g. if no eGFP signal was detected in the MCL of the OB). Investigators performing image analysis and quantification were blinded to the genotype and/or experimental group. For Ribo-TRAP pull-down experiments, all the samples presenting enrichment of the wrong marker genes were excluded. For the quantification of RBM20 expression in the olfactory bulb, statistical analysis was performed with Prism 9 (GraphPad software) using unpaired t-test. Data presented are mean ± SD. Images were assembled using Fiji, Omero (*Swedlow et al., 2003*), and Adobe Illustrator software.

A detailed description of the exclusion criteria for different experiments is included in the respective Materials and methods sections. Statistical analyses were conducted with GraphPad Prism 9. The applied statistical tests were chosen based on sample size, normality of data distribution, and number of groups compared. Details on n-numbers, p-values, and specific tests are found in the figure legends.

## Acknowledgements

We thank Caroline Bornmann and Sabrina Innocenti for excellent support with lab organization and experiments, Pawel Pelzcar and the Centre for Transgenic Models at the University of Basel for outstanding advice and services, Geoffrey Fucile (SciCORE) for help with data analysis, the Biozentrum

Imaging Core Facility for support with image acquisition and analysis, the Biozentrum Proteomic Core Facility for support with the mass spectrometry data analysis, the Quantitative Genomics Centre of the University of Basel for excellent technical assistance. The Scheiffele Laboratory is an associate member of the NCCR RNA & Disease, funded by the Swiss National Science Foundation. This work was financially supported by funds to PS from the Canton Basel-Stadt/University of Basel, the Swiss National Science Foundation (project 179432), a Swiss National Science Foundation Advanced Grant (TMAG-3-209273), and a European Research Council Advanced Grant (SPLICECODE).

## Additional information

### Funding

| Funder | Grant reference number | Author |
|---|---|---|
| European Molecular Biology Organization | Long-term Fellowship | Raúl Ortiz |
| Canton Basel-Stadt/ University of Basel | Associate Member | Peter Scheiffele |
| Swiss National Science Foundation | project 179432 | Peter Scheiffele |
| Swiss National Science Foundation | TMAG-3-209273 | Peter Scheiffele |
| European Research Council | SPLICECODE | Peter Scheiffele |

The funders had no role in study design, data collection and interpretation, or the decision to submit the work for publication.

### Author contributions

Giulia Di Bartolomei, Conceptualization, Data curation, Formal analysis, Validation, Investigation, Visualization, Methodology, Writing – review and editing; Raúl Ortiz, Data curation, Formal analysis, Methodology, Writing – review and editing; Dietmar Schreiner, Data curation, Formal analysis, Validation, Investigation, Visualization, Methodology, Writing – review and editing; Susanne Falkner, Visualization, Methodology; Esther EJM Creemers, Resources, Writing – review and editing; Peter Scheiffele, Conceptualization, Funding acquisition, Investigation, Writing - original draft, Project administration

### Author ORCIDs

Giulia Di Bartolomei ⑤ https://orcid.org/0000-0002-9499-8427
Esther EJM Creemers ⑤ https://orcid.org/0000-0002-9001-3854
Peter Scheiffele ⑤ https://orcid.org/0000-0002-9516-9399

### Ethics

All procedures involving animals were approved by and performed in accordance with the guidelines of the Kantonales Veterinäramt Basel-Stadt (licenses 2599, 2447, 2272) and studies are optimized with regards to the 3R principles. All of the animals were handled according to approved animal use protocols. All surgery was performed under isoflurane anesthesia and every effort was made to minimize suffering of the animals.

Reviewer #1 (Public review): https://doi.org/10.7554/eLife.104808.3.sa1
Reviewer #2 (Public review): https://doi.org/10.7554/eLife.104808.3.sa2
Reviewer #3 (Public review): https://doi.org/10.7554/eLife.104808.3.sa3
Author response https://doi.org/10.7554/eLife.104808.3.sa4

## Additional files

### Supplementary files

Supplementary file 1. Identified RBM20-binding sites in the heart and olfactory bulb tissues. List of RBM20 peaks identified on transcript mRNAs in the heart and in olfactory bulb tissues. Peaks were identified through the peak caller Clipper followed by irreproducible discovery rate (IDR) analysis between replicates. In this table, beyond the standard output parameters produced by IDR, columns containing information about the annotation of the targeted transcript and the position of RBM20-binding site in relation to the exon-intron boundaries (R package 'AnnotatR') are reported. Moreover, the list of read counts summarized using featureCounts for identification of expressed genes in the input samples of heart and olfactory bulb is reported.

Supplementary file 2. Gene ontology analysis of transcripts directly bound by RBM20. Gene ontology analysis results by Panther of mRNA transcripts directly bound by RBM20 RNA-binding protein in both heart and olfactory bulb tissues.

Supplementary file 3. Expressed genes and percentage of mapped and unmapped unique reads in Ribo-TRAP RNA-sequencing experiments. Read counts summarized using featureCounts. For each sample from either *Slc17a6*+ neurons of the olfactory bulb or PV+ interneurons of the neocortex.

Supplementary file 4. Summary of differential gene expression analysis. Expression values (FPKM) of genes identified in Parvalbumin-positive and *Slc17a6*+ neurons in Ribo-TRAP RNA-sequencing experiments and the results of their differential gene expression analysis by DESEQ2 in wild-type vs. *Rbm20* conditional knockout mice.

Supplementary file 5. Summary of alternative exon usage analysis. Expression values (RPKM) of all the exons identified in Parvalbumin-positive and Slc17a6-positive cells in Ribo-TRAP RNA-sequencing experiments and the results of the alternative exon usage analysis in wild-type vs. *Rbm20* conditional knockout mice.

Supplementary file 6. Gene ontology of deregulated transcripts and alternatively spliced exons. Gene ontology analysis results by Panther for deregulated transcripts (all, downregulated, and upregulated) and exon usage analysis in olfactory bulb neurons upon RBM20 ablation.

Supplementary file 7. Analysis of intron length. Analysis of intron length in deregulated, nonregulated, and RBM20-bound transcripts in olfactory bulb neurons.

MDAR checklist

### Data availability

The datasets produced in this study are available in the following databases: MassIVE (code: MSV000093344), PRIDE (code: PXD046806), and GEO (code:GSE250100).

The following datasets were generated:

| Author(s) | Year | Dataset title | Dataset URL | Database and Identifier |
|---|---|---|---|---|
| Di Bartolomei G, Ortiz R, Schreiner D, Falkner S, Creemers EE, Scheiffele P | 2023 | The dilated cardiomyopathy-associated RNA Binding Motif Protein 20 regulates long pre-mRNAs in neurons | https://www.ncbi.nlm.nih.gov/geo/query/acc.cgi?acc=GSE250100 | NCBI Gene Expression Omnibus, GSE250100 |
| Di Bartolomei G, Ortiz R, Schreiner D, Falkner S, Creemers EE, Scheiffele P | 2023 | The dilated cardiomyopathy-associated RNA Binding Motif Protein 20 regulates long pre-mRNAs in neurons | https://massive.ucsd.edu/ProteoSAFe/dataset.jsp?task=ed356ca6c5ff4f3883b0449bf4b3b5a5 | MassIVE, MSV000093344 |
| Di Bartolomei G, Ortiz R, Schreiner D, Falkner S, Creemers EE, Scheiffele P | 2023 | The dilated cardiomyopathy-associated RNA Binding Motif Protein 20 regulates long pre-mRNAs in neurons | https://proteomecentral.proteomexchange.org/cgi/GetDataset?ID=PXD046806 | ProteomeXchange, PXD046806 |

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
