## [Editor Report · eLife Assessment]

This study reports that the RNA binding and cardiomyopathy-associated protein RBM20 is expressed in specific populations of neurons in the CNS, where it binds to and regulates the expression of synapse-related RNAs. This is an **important** finding because it reveals a new mechanism for gene regulation in neurons by an RNA binding protein previously studied in the heart; the authors also provide data to suggest that the mechanism by which RBM20 acts in neurons may be distinct from the splicing regulation studied in cardiac tissue. The data in support of the binding and regulation of RNAs by RBM20 is **compelling**, using leading edge sequencing methods to determine RNA binding profiles, and cell type specific genetics for evaluation of function.

---

## [Referee Report · Reviewer #1 (Public review)]

Summary:

The authors of this study set out to find RNA binding proteins in the CNS in cell-type specific sequencing data and discover that the cardiomyopathy-associated protein RBM20 is selectively expressed in olfactory bulb glutamatergic neurons and PV+ GABAergic neurons. They make an HA-tagged RBM20 allele to perform CLIP-seq to identify RBM20 binding sites and find direct targets of RBM20 in olfactory bulb glutmatergic neurons. In these neurons, RBM20 binds intronic regions. RBM20 has previously been implicated in splicing, but when they selectively knockout RBM20 in glutamatergic neurons they do not see changes in splicing, but they do see changes in RNA abundance, especially of long genes with many introns, which are enriched for synapse-associated functions. These data show that RBM20 has important functions in gene regulation in neurons, which was previously unknown, and they suggest it acts through a mechanism distinct from what has been studied before in cardiomyocytes.

Strengths:

The study finds expression of the cardiomyopathy-associated RNA binding protein RBM20 in specific neurons in the brain, opening new windows into its potential functions there.

The study uses CLIP-seq to identify RBM20 binding RNAs in olfactory bulb neurons.

Conditional knockout of RBM20 in glutamatergic or PV neurons allows the authors to detect mRNA expression that is regulated by RBM20.

The data include substantial controls and quality control information to support the rigor of the findings.

Weaknesses:

The authors do not fully identify the mechanism by which RBM20 acts to regulate RNA expression in neurons, though they do provide data suggesting that neuronal RBM20 does not regulate alternate splicing in neurons, which is an interesting contrast to its proposed mechanism of function in cardiomyocytes. Discovery of the RNA regulatory functions of RBM20 in neurons is left as a question for future studies.

The study does not identify functional consequences of the RNA changes in the conditional knockout cells, so this is also a question for the future.

---

## [Referee Report · Reviewer #2 (Public review)]

Summary:

The group around Prof. Scheiffele has made seminal discoveries reg. alternative splicing that is reflected by a current ERC advanced grant and landmark papers in eLife (2015), Science (2016), and Nature Neuroscience (2019). Recently, the group investigated proteins that contain an RRM motif in the mouse cortex. One of them, termed RBM20, was originally thought be muscle-specific and involved in alternative splicing in cardiomyocytes. However, upon close inspection, RBP20 is expressed in a particular set of interneurons (PV positive cells of the somatosensory cortex) in the cortex as well as in mitral cells of the olfactory bulb (OB). Importantly, they used CLIP to identify targets in the OB and heart. Next and quite importantly, they generated a knock-in mouse line with a His-biotin acceptor peptide and a HA epitope to perform specific biochemistry. Not surprisingly, this allowed them to specifically identify transcripts with long introns, however, most of the intronic binding sites were very distant to the splice sites. Closer GO term inspection revealed that RBM20 specifically regulates synapse-related transcripts. In order to get in vivo insight into its function in the brain, the authors generated both global as well as conditional KO mice. Surprisingly, there were no significant differences in in RBM20 PV interneurons, however, 409 transcripts were deregulated in in OB glutamatergic neurons. Here, CLIP sites were mostly found to be very distant from differentially expressed exons. Furthermore, loss-of-function RBM20 primarily yields loss of transcripts, whereas upregulation appears to be indirect. Together, these results strongly suggest a role of RBM20 in the inclusion of cryptic exons thereby promoting target degradation.

Strengths:

The quality of the data and the figures is high, impressive and convincing. The reported results strongly suggest a role of RBM20 in the inclusion of cryptic exons thereby promoting target degradation.

Weaknesses:

In their revised manuscript, the authors significantly improved the intro and results section, which is now much better suited for the general public and allows better to follow the logic of the experiments. Also, the discussion has now been expanded doing better justice to the importance of the findings presented.

In my opinion, the revised manuscript clearly improved and represents a timely and important study, which provides major new insight into the expression and possible function of RBM20 in tissues outside of muscle.

---

## [Referee Report · Reviewer #3 (Public review)]

Summary:

The authors identified RBM20 expression in neural tissues using cell type-specific transcriptomic analysis. This discovery was further validated through in vitro and in vivo approaches, including RNA fluorescent in situ hybridization (FISH), open-source datasets, immunostaining, western blotting, and gene-edited RBM20 knockout (KO) mice. CLIP-seq and RiboTRAP data demonstrated that RBM20 regulates common targets in both neural and cardiac tissues, while also modulating tissue-specific targets. Furthermore, the study revealed that neuronal RBM20 governs long pre-mRNAs encoding synaptic proteins.

Strengths:

• Utilization of a large dataset combined with experimental evidence to identify and validate RBM20 expression in neural tissues.

• Global and tissue-specific RBM20 KO mouse models provide robust support for RBM20 localization and expression.

• Employing heart tissue as a control highlights the unique findings in neural tissues.

Weaknesses:

• Lack of physiological functional studies to explore RBM20's role in neural tissues.

• Data quality requires improvement for stronger conclusions.

Comments on revisions:

The authors have effectively addressed most of my concerns, which has significantly improved the quality and reliability of the data. While sufficient functional data were not provided, the current findings offer valuable and novel insights into the expression of RBM20 in neurons. I have no further concerns.

---

## [Author Response]

The following is the authors’ response to the original reviews.

**Public reviews:**

We thank the three reviewers for the constructive suggestions made in the Public Reviews and the Recommendations to Authors. We have now addressed these comments in a revised manuscript as follows:

(1) We will revise the text according to the reviewer suggestions and provide more detailed explanations in results and discussion.

(2) We have uploaded higher resolution images of several figures (resolution had been reduced to achieve lower file sizes) to address the comment regarding “data quality”.

(3) We have included additional data on eCLIP control experiments in the supplementary figures.

(4) We have performed additional replications of the western blot analysis for Rbm20 knock-out animals and provided the data in a new Figure.

**Recommendations for the authors:**

**Reviewer #1:**
(1) The study is missing CLIP-seq data from control mice that do not express HA, or HA-knocked into a safe-harbor locus. This is important because there is plenty of background HA staining in Figure S2B, in wild-type mice. Including this control would allow subsequent peak calling to distinguish between non-specific HA peaks and RBM20 specific peaks.

The biochemical conditions used in immunostaining are much less stringent than the buffers employed for immunoprecipitation in the eCLIP protocol. Thus, background staining is not a an informative reference to assess specificity of CLIP isolations. In previous experiments, we confirmed very low background with the anti-HA antibodies in our eCLIP protocol. In the present study, we used a “no-crosslinking control” where samples were not irradiated with UV light. This negative control is now included in Supplementary Figure 4.

(2) The GO analysis performed to infer synapse-gene specific regulation would be more useful if the authors would discuss specific genes that are represented within these terms and have been shown to be associated with neuronal function.

We have now noted several synapse-related genes identified in the text.

(3) Some figures would benefit from larger size and higher resolution including Fig S1, S3.

We had previously embedded Figures as png files in the text document. In the revised version we uploaded the figures in higher resolution as individual jpeg files. Moreover, we now split Figure S1 into two separate supplementary figures (new Fig.S2) which allowed for enlarging the size of panels. We further enlarged the panels of (former) Fig.S3 (now Fig.S4).

(4) RBP genes in Figure 1A x-axis are all lowercase. This is not standard mouse gene nomenclature.

We corrected this.

(5) Typo in Figure S4F rightmost panel y-axis - 'Length' is misspelled.

We corrected this.

**Reviewer #2:**
Minor points:- Shortly explain DESEQ2 (p4)

We now added a brief note and corresponding reference in the main text of the manuscript.

- Is RBM20 a shuttling protein? Any detection in the cytoplasm?

Our immunostainings for the endogenous RBM20 in heart and olfactory bulb cells suggest that the vast majority of wild-type RBM20 is localized to the nucleus. Previous work on RBM20 disease mutants suggest that pathological forms can accumulate in the cytoplasm. However, with the sensitivity of our detection we did not obtain evidence for a significant cytoplasmic pool in neurons. This does not exclude the possibility that the protein is shuttling – but assessing this would require different types of experiments.

**Reviewer #3:**
(1) Figure 1C: It is shown that some of the RBM20 staining do not colocalize with PV. This observation requires further explanation and discussion to clarify the significance.

As seen in the fluorescent in situ hybridizations as well as the RiboTRap purifications (Fig.S1C,D), we observe mRNA RBM20 expression not only in parvalbumin-positive interneurons but also somatostatin-positive cells of the neocortex. Accordingly, some RBM20-positive cells do not express parvalbumin. We now clarified this in the text.

Additionally, in Figure S1C, the resolution of the image is low, making it difficult to conclusively determine whether RBM20 RNA is localized in the nucleus. A high-resolution image would be beneficial to address this ambiguity.

The Rbm20 mRNA is localized in the nucleus and cytoplasm. We have now split Figure S1 into two separate figures to enlarge the panels for S1C and make this more visible. Moreover, we uploaded higher resolution figure files.

(2) Figure 1E: The molecular weight of RBM20 is approximately 135 kDa, yet there is a band near 135 kDa in the KO heart. How do the authors determine that the 150 kDa band represents RBM20 rather than the 135 kDa band? The authors may consider increasing the sample size to confirm whether the smaller band consistently appears across all KO heart tissues.

We appreciate that in this higher molecular weight range, the indicated weight markers may not be entirely accurate. We used a validated knock-out mouse line to identify the appropriate RBM20 protein band. As the 150kDa band was reproducibly lost in the knock-out tissue in the brain and the heart tissue whereas the fainter band of lower mobility remained we concluded that on our gel system RBM20 protein has an apparent molecular weight of 150 kDa. This is further supported by the fact that also the endogenously tagged RBM20 protein has a similar mobility.

As suggested by the reviewer, we now re-ran Western blots from multiple wild-type and corresponding knock-out tissues. This further confirmed the migration of the protein and loss of the 150 kDa band in the mutant mice (new Figure 1E).

(3) Figure 2A: A higher-resolution image is recommended. Prior studies on RBM20 mutation knock-in mice suggest that when RBM20 localizes to the cytoplasm, it promotes molecular condensate formation. This seems to be the case in Figure 2A; however, the low image quality makes it difficult to see these molecular condensates.

Figure2A shows endogenous RBM20 (not the epitope-tagged protein in the knock-in mice). The vast majority of the protein is localized in the nucleus rather than the cytoplasm. We are a bit uncertain what “condensates” the reviewer refers to. In the heart, we indeed see accumulations of RBM20 in foci (as described previously in the literature). As judged by their location within the DAPI-positive area, these foci are in the nucleus. By contrast, in the olfactory bulb neurons (which express lower levels of RBM20) we do not see a comparable concentration in nuclear foci but rather broad and diffuse staining. This is consistent with the hypothesis that the nuclear foci depend on the expression of highly expressed target transcripts such as titin. To better visualize this, we now uploaded files with higher resolution for the revised manuscript.

(4) Figure 4D: This figure is not cited in the main text and should be referenced appropriately.

We corrected this.

(5) Page 5: The sentence "Finally, introns bound by RBM20 were significantly longer than expected by chance as assed..." contains a typo. The word "assed" should be corrected to "assessed".

We corrected this.

(6) Functional data: The study would benefit from functional experiments to elucidate the physiological role of RBM20 in PV neurons. For instance, since RBM20 regulates calcium-handling genes in neurons, does its absence impair calcium signaling in PV neurons? Additionally, given that RBM20 is involved in synaptic regulation, could RBM20 KO disrupt synaptic function? While it may not be feasible to address all these questions, providing some functional data would greatly enhance the overall significance of the study.

We completely agree with the reviewer that this would greatly advance the study and the lack of data on cellular functions is the most significant limitation of this work. We attempted to obtain insights into cellular function through the structural investigations (Fig.S5). We had obtained some data on a behavioral phenotype in the mice which indicates that knock-out in vGLUT2 neurons precipitates alterations in behavior. However, due to conditions in our animal facility (emissions from construction) we struggled to solidify/confirm this data. Thus, in the interest of sharing the existing data in a timely manner we felt that more elaborate functional studies on synaptic transmission or calcium imaging should better be performed in a separate effort.